# Learning with Real-time Improving Predictions in Online MDPs

## Abstract

In this paper, we introduce the Decoupling Optimistic Online Mirror Descent (DOOMD) algorithm, a novel online learning approach designed for episodic Markov Decision Processes with real-time improving predictions. Unlike conventional methods that employ a fixed policy throughout each episode, our approach allows for continuous updates of both predictions and policies within an episode. To achieve this, the DOOMD algorithm decomposes decision-making across states, enabling each state to execute an individual sub-algorithm that considers both immediate and long-term effects on future decisions. We theoretically establish a sub-linear regret bound for the algorithm, providing a guarantee on the worst-case performance.

## 1 Introduction

In this paper, we study the problem of online episodic Markov decision processes (MDPs) with real-time improving predictions. A learner interacts with an environment over $T$ episodes, each of a finite length. During each episode, the learner operates within an MDP – selects actions based on observed states, incurs a cost [1], and transitions to subsequent states. Before making each action, the learner has access to external predictions for future steps. These predictions, while imperfect, can facilitate decision-making and are dynamically updated in real time as the learner interacts with the environment. Importantly, these predictions are expected to become more accurate as the episode progresses.

Consider the example of routing, where over 93% of travelers rely on GPS navigation like Google Maps (CarPro, 2022). These tools use historical data and machine learning algorithms to forecast future traffic conditions and estimate travel time, updating predictions in real-time as one progresses along a route (Derrow-Pinion et al., 2021). Typically, predictions tend to become more accurate as the destination approaches, since there is less need for forecasting distant events. Due to this trend, trivially trusting initial predictions may not be a good strategy. For instance, consider a traveler moving from Node 1 to Node 4 in Figure 1. The traveler initially selects the route $1 \rightarrow 2 \rightarrow 3$ based on an early prediction. However, upon arriving at Node 2, a more accurate prediction indicates that the chosen route is always the worst no matter what decision is made here.

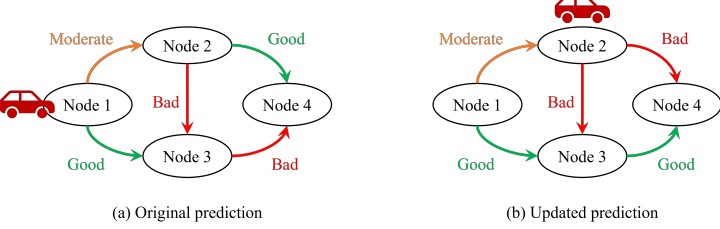

(a) Original prediction                    (b) Updated prediction

Figure 1: A motivating example for decisions under predictions

Real-time improving predictions are becoming increasingly prevalent, especially in an era where predictive capabilities are rapidly advancing due to machine learning breakthroughs (Agrawal et al.,

---

[1]We use costs throughout the paper, which is equivalent to negative rewards

2022) and the development of Large Language Models (Wang et al., 2023; Jablonka et al., 2024). Examples range from self-driving cars, which rely on predictions of other vehicles' trajectories (Cao et al., 2023), to resource allocation strategies that depend on forecasts of future demand (Lei et al., 2020). A common pattern across these applications is that predictions for distant events tend to be less accurate than those for the near future, with predictions improving as the learner approaches the end of an episode.

Despite the trend, improving predictions do not guarantee better outcomes, as demonstrated in the earlier routing example. The traveler, following a greedy policy, fails to benefit from the updated predictions. Indeed, we need a decision framework to exploit the increasing accuracy, which raises fundamental questions: How much trust should we place in each prediction? How can we leverage predictions to update policies dynamically? Can we still maintain a performance guarantee?

Conventional online learning algorithms in episodic MDPs (Neu et al., 2012; Dick et al., 2014; Rosenberg & Mansour, 2019a;b; Jin et al., 2020; Shani et al., 2020; Neu & Pike-Burke, 2020; Cai et al., 2020; Rosenberg et al., 2020; Mao et al., 2021; Neu & Olkhovskaya, 2021; Jin et al., 2021) fail short in addressing these questions. These algorithms typically *treat the policy within each episode as fixed, and only update it between episodes*. While a few works (Cai et al., 2020; Neu & Olkhovskaya, 2021) explore updating policies within episodes, these updates are usually done for computational convenience and can be reformulated into an equivalent approach with only between-episode updates. Existing approaches to leveraging predictions in episodic MDPs have generally assumed that the learner updates their policy based on predictions at the start of each episode, with no further changes made during the episode (Rakhlin & Sridharan, 2013; Steinhardt & Liang, 2014; Guan, 2015; Fei et al., 2020). However, leveraging real-time improving predictions requires the learner to *continuously update the policy during episode*, which goes beyond the existing frameworks.

To handle these dynamic updates effectively, policies need to be decomposed into meaningful decision components. This decomposition enables the learner to update future components of the policy while executing the current decisions. Intuitively, it helps the learner focus more on the present, and defer later decisions until more accurate information becomes available. For instance, in Figure 1, when the travel is at Node 1, the decision should focus only on choosing between Node 2 and 3, without considering decisions afterward. However, quantifying the contribution of each decision to the overall performance is not straightforward. Each decision not only has immediate effects but also influences the long-term trajectory by shaping the remaining decision space. In the motivating example, the poor decision at Node 1 restricts the available options at Node 2, leaving the traveler with only suboptimal routes. Our goal is to systematically decompose the decision and develop algorithms that can update each decision based on real-time predictions.

**Contribution.** In this paper, we propose a novel framework that allows a more general and flexible interaction between the learner and the environment in online episodic MDPs. Unlike conventional approaches that employ a fixed policy per episode, our approach allows for continuous updates of both predictions and policies within an episode. To achieve this, we decompose the policy to every state and introduce a concept of *cumulative cost*, which accounts for both immediate costs and the long-term impact on future decisions. Using this concept, we propose the *Decoupling Optimistic Online Mirror Descent (DOOMD) algorithm*, which implements sub-algorithms at each state, aiming to control its total cumulative cost over time.

This paper makes the following contributions. (1) We provide a systematic model for online MDPs that allows dynamic policy updates within an episode to accommodate improving predictions. To the best of our knowledge, this problem has never been explored in the literature. (2) By utilizing cumulative costs, we prove that the total regret can be decomposed to reflect each decision's actual contribution to the overall performance. (3) We prove that the DOOMD algorithm achieves a sublinear regret bound of $O(\sqrt{T})$, both for fixed and dynamically updating learning rates.

**Organization.** The remainder of this paper is structured as follows. Section 2 discusses related works. Section 3 introduces the model. Section 4 presents the online algorithm and Section 5 analyzes its regret bound. Lastly, Section 6 concludes the paper.

**Notations.** For a positive integer $n$, denote $[n] = \{0, 1, .., n\}$. For $n$ sets, $\mathcal{X}^1, ..., \mathcal{X}^n$, denote $\mathcal{X}^{1:n} = \cup_{k=1}^n \mathcal{X}^k$. The inner product of two vectors $a, b$ is denoted as $\langle a, b \rangle = a^T b$. A comprehensive table

of notations is provided in Appendix A. Throughout the paper, unless otherwise specified, proofs are provided in Appendix E.

## 2 RELATED WORKS

### 2.1 ONLINE LEARNING IN MDPs

Many real-world optimization and control problems can be modeled using online episodic MDPs, including routing (Choudhury et al., 2019; Wu et al., 2022), finance (Hambly et al., 2023), scheduling (Yin et al., 2020), and self-driving (Cao et al., 2023). Given the unknown nature of future steps or episodes, the learner needs to learn while interacting with the environment. The objective is typically to minimize the regret, which is defined as the difference between the learner's cumulative cost and that of an optimal policy, denoted as $\mathbf{R}_T = \sum_{t=1}^{T} C_t(\pi_t) - \sum_{t=1}^{T} C_t(\pi_t^*)$, where $C_t(\pi_t)$ represents the expected cost under policy $\pi_t$ in episode $t$, and $\pi_t^*$ refers to a hindsight optimal policy. Prior research has extensively explored online learning algorithms for episodic MDPs, with various settings for rewards and transitions (stochastic or adversarial) and information states (full or bandit feedback), including Neu et al. (2012); Dick et al. (2014); Rosenberg & Mansour (2019a;b); Jin et al. (2020); Mao et al. (2021); Neu & Olkhovskaya (2021); Jin et al. (2021); Shani et al. (2020); Neu & Pike-Burke (2020); Cai et al. (2020); Rosenberg et al. (2020), among others.

As noted earlier, most existing approaches often treat the policy within each episode as fixed and static. Viewing online learning in episodic MDPs through this lens connects the problem to a broader class of online optimization methods, where each episode is akin to making a single decision (Chiang et al., 2012; Wei & Zhang, 2020; Bhaskara et al., 2020; Jiang et al., 2023). We refer the readers to Orabona (2019) for a comprehensive introduction to online optimization. In fact, using occupancy measures to represent policies, online episodic MDPSs can be transformed into an equivalent online linear optimization problem (Zimin & Neu, 2013).

Another closely related research direction is online learning in non-stationary MDPs. In such scenarios, the learner interacts with the environment continuously for $T$ steps rather than in episodic structures. Starting from the pioneering works by Even-Dar et al. (2004; 2009), this problem has been extensively studied under various settings (Joulani et al., 2013; Neu et al., 2010; Neu & Gómez, 2017; Li et al., 2019b; Lecarpentier & Rachelson, 2019; Rivera Cardoso et al., 2019; Cheung et al., 2020; Chandak et al., 2020; Cheung et al., 2023). These studies focus on adapting policies to account for shifts in the non-stationary environment over time.

### 2.2 UTILIZATION OF PREDICTIONS

From the theoretical perspective, predictions can act as a form of regularity assumption, similar to conditions like Lipscitz continuity (Lecarpentier & Rachelson, 2019) or total variation (Cheung et al., 2020; 2023). Such assumptions can mitigate the conservativeness of the algorithms designed for adversarial MDPs. As online learning algorithms utilizing predictions in episodic MDPs still assume the learner carries out a fixed policy per episode (Rakhlin & Sridharan, 2013; Steinhardt & Liang, 2014; Guan, 2015; Fei et al., 2020), these approaches can be linked to online optimization under predictions (Chen, 2018; Purohit et al., 2018; Li et al., 2019a; Li & Li, 2020; Christianson et al., 2022) in a similar way.

Other relevant frameworks to utilize predictions are the predict-then-optimize (Wang et al., 2021; Elmachtoub & Grigas, 2022) and performative prediction (Perdomo et al., 2020) which integrates the training of predictive models with decision optimization. However, these approaches differ fundamentally from ours, as we focus on leveraging exogenous predictions. This is especially useful in real-world applications, as generating accurate in-house predictions may be impractical due to constraints on time and resources. For instance, in routing, most users rely on external systems like Google Maps to predict travel times rather than generating their own predictions. Additionally, another line of research explores whether and when to stop updating predictions (Lee et al., 2024), which also pursues a distinct objective from ours.

## 3 MODEL

### 3.1 ONLINE MDP

The episodic MDP is defined by the tuple $\{\mathcal{X}, \mathcal{A}, P, c\}$, where $\mathcal{X}$ denotes the state space, $\mathcal{A}$ denotes the action space, $P : \mathcal{X} \times \mathcal{A} \times \mathcal{X} \to \mathbb{R}$ and $c : \mathcal{X} \times \mathcal{A} \to \mathbb{R}$ represent the transition and cost functions, respectively. We consider a fixed and known transition function $P$, while the cost function may vary across episodes. The cost function for episode $t$ is denoted as $c_t$, and for simplicity, it is normalized to $[0, 1]$. We also denote the set of all episodes as $\mathcal{T} = \{1, ..., T\}$.

Without losing generality, we assume the state space $\mathcal{X}$ follows a layered structure, forming a loop-free episodic MDP. Specifically, the state space is partitioned into $L + 1$ layers $\mathcal{X} = \cup_{l \in \mathcal{L}} \mathcal{X}^l$, with $\mathcal{L} = \{0, ..., L\}$, and the initial layer $\mathcal{X}^0$ only contains a single state $x^0$. This assumption is not restrictive as any episodic MDP can be reconstructed as an equivalent loop-free structure (Maran et al., 2023).

In addition, we simplify the transition function $P$ for clarity to a deterministic function where $P(x'|x, a) = 1$ if and only if $x' = a$. This assumption is not fundamental to our analysis and can be easily generalized to stochastic transitions, which is detailed in Appendix C. Under this setup, the action space for state $x^l \in \mathcal{X}^l$ links directly to the states on the subsequent layer, i.e. $\mathcal{A}(x^l) \subseteq \mathcal{X}^{l+1}$. We define the state-action pairs on layer $l$ as $\mathcal{U}^l = \{(x, a) : x \in \mathcal{X}^l, a \in \mathcal{A}(x)\}$, and the set of all state-action pairs as $\mathcal{U} = \mathcal{U}^{0:L-1}$. Consequently, the cost function in any episode $t$ can be viewed as $c_t : \mathcal{U} \to [0, 1]$.

The interaction between the learner and the environment follows the protocol outlined as follows. For each episode $t$, after reaching layer $l$, the learner receives an updated cost prediction for state-action pairs in all subsequent layers, denoted as $M_t^l : \mathcal{U}^{l:L-1} \to [0, 1]$. For each state-action pair $u \in \mathcal{U}^k$ (with $k \geq l$), the predicted cost is denoted as $M_t^l(u)$. The accuracy of the prediction is characterized by the error bound $\epsilon^l$, such that $|M_t^l(u) - c_t(u)| \leq \epsilon^l, \forall t \in \mathcal{T}, k \geq l, u \in \mathcal{U}^k$. These real-time predictions enable the learner to update their policy at each layer dynamically. Let $\pi_t^l : \mathcal{X}^l \times \mathcal{A}(\mathcal{X}_l) \to [0, 1]$ denote the policy used at layer $l$ in episode $t$. At the end of each episode, the learner receives full information regarding the cost function $c_t$.

In addition, we assume that as the learner continues interacting with the environment, the uncertainty decreases, resulting in gradually improving predictions. Otherwise, it makes no sense to update the policy based on predictions. Given an exogenous prediction sequence $\boldsymbol{M} = \left\{M_t^l\right\}_{t \in \mathcal{T}, l \in \mathcal{L}}$, a learning algorithm generates a set of policies $\boldsymbol{\pi} = \left\{\pi_t^l\right\}_{t \in \mathcal{T}, l \in \mathcal{L}}$, which induces an expected total cost $C_T(\boldsymbol{\pi}) = E\left[\sum_{t=1}^{T} \sum_{l=0}^{L-1} c_t(x_t^l, a_t^l) \middle| \boldsymbol{M}, \boldsymbol{\pi}\right]$, where $E[\cdot | \boldsymbol{M}, \boldsymbol{\pi}]$ indicates that the state and action on each layer ($x_t^l$ and $a_t^l$) are generated by policy $\boldsymbol{\pi}$ under predictions $\boldsymbol{M}$. By selecting the optimal stationary policy in hindsight as the baseline, the regret of the algorithm is defined as $\mathbf{R}_T = C_T(\boldsymbol{\pi}) - \min_{\boldsymbol{\pi}^*} C_T(\boldsymbol{\pi}^*)$, where the minimum is taken over all the stationary policies, i.e., $\pi_t(a|x) = \pi_{t'}(a|x)$ for all $t, t' \in \mathcal{T}, x \in \mathcal{X}$ and $a \in \mathcal{A}(x)$, which captures the opportunity loss from not employing the optimal strategy (Taherkhani et al., 2021). This concept of static regret is commonly adopted in the literature, as in Zimin & Neu (2013); Dick et al. (2014), etc. Our goal is to design a robust algorithm that guarantees a sublinear regret bound (e.g. $O(\log T), O(\sqrt{T})$), so that, on average, the algorithm performs as well as the best stationary policy when $T$ is large.

### 3.2 ONLINE LINEAR OPTIMIZATION

Building on existing studies that employ occupancy measures to design algorithms for online MDPs (Zimin & Neu, 2013; Dick et al., 2014; Zhao et al., 2022), we adopt a similar approach to streamline our framework. The occupancy measure induced by a policy in an episode $\pi = \left\{\pi^l\right\}_{l \in \mathcal{L}}$ is denoted as $w^\pi \in K \subseteq [0, 1]^{|\mathcal{U}|}$, which represents the probability of executing each state-action pair under the policy $\pi$. The domain of occupancy measure is defined by the set:

$$K = \left\{w : \sum_{u \in \mathcal{U}^l} w(u) = 1, \sum_{a \in \mathcal{A}(x)} w(x, a) = \sum_{x' \in \mathcal{A}^{-1}(x)} w(x', x), \forall l \in [L-1], x \in \mathcal{X}^{l+1}\right\}, \quad (1)$$

where the first condition ensures the occupancy measure on each layer forms a valid distribution and the second corresponds to the flow conservation equation between layers. $\mathcal{A}^{-1}(x)$ represents the set of preceding states $\{x' \in \mathcal{X} : x \in \mathcal{A}(x')\}$.

Occupancy measures effectively translate MDP policies into an equivalent but more tractable form. Given policy $\pi$, we can recursively compute its induced occupancy measure starting from layer 0. Conversely, a policy for each layer can be reconstructed from an occupancy measure $w$ by $\pi^w(a|x) = \frac{w(x,a)}{\sum_{a' \in \mathcal{A}(x)} w(x,a')}$ $a \in \mathcal{A}(x)$. Therefore, finding the optimal policy is equivalent to finding the optimal occupancy measure. With slight abuse of notation, express the cost as a vector $c \in [0,1]^{|\mathcal{U}|}$, then the expected total cost introduced by policy $\pi_t$ in episode $t$ is $\langle c_t, w^{\pi_t} \rangle$. Hence, the cumulative regret over $T$ episodes becomes $\mathbf{R}_T = \sum_{t=1}^{T} \langle c_t, w^{\pi_t} \rangle - \min_{w \in K} \sum_{t=1}^{T} \langle c_t, w \rangle$.

## 4 ALGORITHM

We now introduce an algorithm specifically designed to exploit the layered structure, accommodating dynamic predictions while aiming to achieve a sub-linear regret bound.

### 4.1 AN ILLUSTRATIVE EXAMPLE

To better present the algorithm, let us start with a motivating example involving five states distributed across three layers, as shown in Figure 2. The technical insight is that the total algorithm regret can be decomposed into contributions from individual states, each of which can be effectively managed.

**Decision decomposition.** In our setup, decision-making is decentralized to individual states. For instance, as indicated by the colors in the figure, at the initial state $x_0$, decisions are only concerned with transitions $(x_0, x_1)$ and $(x_0, x_2)$, without considering subsequent states. This localized approach results in maintaining three distinct occupancy measures: $w_t^0$ for $\mathcal{U}^0$, and $w_t^1$, $w_t^2$ for $\mathcal{U}^1(x_1)$ and $\mathcal{U}^1(x_2)$ respectively, where $\mathcal{U}^l(x) = \{(x,a) : a \in \mathcal{A}(x)\}$ for $x \in \mathcal{X}^l$. For simplicity, write the state-action pair as $u_{ij} = (x_i, x_j)$. These occupancy measures satisfy that $w_t^0(u_{01}) + w_t^0(u_{02}) = w_t^1(u_{13}) + w_t^1(u_{14}) = w_t^2(u_{23}) + w_t^2(u_{24}) = 1$.

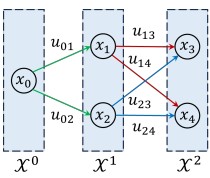

Figure 2: An example with three layers

**Regret decomposition.** Each occupancy measure, such as $w_t^1$, can be interpreted as a conditional probability distribution, depending on reaching $x_1$. Consequently, the actual probability of executing state-action pair $u_{13}$ is the product $w_t^0(u_{01})w_t^1(u_{13})$. Compared with any occupancy measure $w \in K^\delta = \{w \in K : w(u) > \delta, \forall u \in \mathcal{U}\}$, where each state-action pair has a minimum visit probability $\delta > 0$, the cost difference for $u_{13}$ in episode $t$ is:

$$
\begin{aligned}
&c_t(u_{13})[w_t^0(u_{01})w_t^1(u_{13}) - w(u_{13})] \\
=&w_t^1(u_{13})c_t(u_{13})[w_t^0(u_{01}) - w(u_{01})] + [w(u_{13}) + w(u_{14})]c_t(u_{13})\left(w_t^1(u_{13}) - \tilde{w}(u_{13})\right)
\end{aligned} \tag{2}
$$

due to the flow conservation of $w$, where $\tilde{w}(u_{13}) := \frac{w(u_{13})}{w(u_{13}) + w(u_{14})}$.

This equation divides the cost difference into two components: one directly resulting from decisions on state $x_1$ (the latter) and the other influenced by prior choices at layer 0 (the former). Therefore, decisions made on state $x_1$ should focus on minimizing the second component in Equation (2), which can be achieved by implementing optimistic online mirror descent (OOMD) (Rakhlin & Sridharan, 2013) at state $x_1$ as a sub-algorithm, which will be detailed in the next section.

**Cumulative costs.** The first component in the regret equation above and a similar component for the regret on state-action pair $u_{14}$ are associated with the decision on layer 0. Thus, the contribution

of decisions on $u_{01}$ to the overall regret is:

$$c_t(u_{01})[w_t^0(u_{01}) - w(u_{01})] + \left[w_t^1(u_{13})c_t(u_{13}) + w_t^2(u_{14})c_t(u_{14})\right][w_t^0(u_{01}) - w(u_{01})] \tag{3}$$
$$= \tilde{c}_t(u_{01})[w_t^0(u_{01}) - w(u_{01})],$$

where the cumulative cost is defined as $\tilde{c}_t(u_{01}) := c_t(u_{01}) + w_t^1(u_{13})c_t(u_{13}) + w_t^2(u_{14})c_t(u_{14})$. This concept reflects the influence of decisions at earlier layers on the overall performance. Constructing a prediction for $\tilde{c}_t(u_{01})$ and analyzing its accuracy is not trivial, as $w_t^1$ remains undetermined until the next layer. This will be addressed in the next section.

**Regret bound adjustment.** We are missing the last component in the regret, as the optimal occupancy measure $w^*$ should be selected from the entire domain $K$ rather than the restricted one $K^\delta$. To address this discrepancy, the regret associated with state-action pair $u_{13}$ can be bounded by:

$$c_t(u_{13})[w_t^0(u_{01})w^1(u_{13}) - w^*(u_{13})] \le c_t(u_{13})[w_t^0(u_{01})w^1(u_{13}) - w(u_{13})] + \delta, \tag{4}$$

with $w \in K^\delta$. By setting $\delta$ to a sufficiently small value (e.g. $1/T$), the additional term negligibly affects the overall regret order.

## 4.2 GENERAL CASES

Building on the foundational concept introduced earlier, our algorithm, termed Decoupling Optimistic Online Mirror Descent or DOOMD, systematically decomposes decision-making across various layers. This approach ensures that each state independently manages an occupancy measure for its respective state-action pairs.

**Notations.** For convenience, let us first clarify the notations used in the algorithm.

- *Costs and predictions*: For cost $c_t$, denote the cost for state-action pairs related to state $x$ as a vector $c_t(x) = \{c_t(x, a) : a \in \mathcal{A}(x)\}$. The overall prediction received on layer $l$ $M_t^l$ and the prediction related to state $x \in \mathcal{X}^l$, $M_t^l(x)$, follow a similar structure. The cumulative cost and prediction on each layer $l$ are denoted as $\tilde{c}_t^l, \tilde{M}_t^l \in [0,1]^{|\mathcal{U}^l|}$, respectively.

- *Decoupled occupancy measures*: For each state $x$, two occupancy measures, $g_t^l(x)$ and $w_t^l(x)$, are defined over $\mathcal{U}(x)$ – The former is recursively maintained based on prior experiences; the latter is updated using predictions, which will be implemented. For any occupancy measure such as $w_t^l(x)$, denote the probability of choosing action $a$ at state $x$ as $w_t^l(x, a) = w_t^l(x)(a)$. Denote the overall occupancy measures for episode $t$ as $w_t = \left\{w_t^l(x)\right\}_{l=0,...,L-1, x \in \mathcal{X}^l}$, and $g_t$ follows a similar definition.

**Algorithm overview.** As detailed in Algorithm 1, DOOMD operates in two phases: preparation (line 5 to line 10) and execution (line 11 to line 20). During the preparation phase, the algorithm first summarizes previous experiences by computing the cumulative cost $\tilde{c}_{t-1}^l$ of all layer $l$, which is subsequently used in the first-step OOMD update to compute $g_t$. In the execution phase, upon observing the realized state and receiving the update prediction $M_t^l$, the algorithm constructs the cumulative predictions $\tilde{M}_t^l$ and performs a second-step OOMD update to calculate the occupancy measure $w_t^l$, which is subsequently implemented.

**Cumulative costs and their prediction.** This procedure generalizes the method used to calculate the cumulative costs presented in Equation (3) for the illustrative example. Specifically, cumulative costs are computed using a backward iteration process outlined in Algorithm 2 in Appendix B. This algorithm progresses from the terminal layer to the initial layer, where each cumulative cost consists of two components: the direct cost and a weighted average of the costs associated with all state-action pairs in the subsequent layer. The weights for this averaging process are determined by the occupancy measures.

Similarly, Algorithm 3 in Appendix B recursively calculates cumulative predictions from layer $L-1$ to some given layer $l$. Since the occupancy measure $w_t^k$ (with $k > l$) will be updated in the future layer $k$, it is underdetermined at the current layer. Therefore, the other occupancy measure $g_t$, which is already computed based on prior experiences, is used in these calculations

**One-step OOMD update.** A key component of the DOOMD algorithm is the one-step update in each OOMD sub-algorithm (Rakhlin & Sridharan, 2013), which is detailed in Algorithm 4 in Appendix B. This update adjusts the occupancy measure to minimize the incurred costs (either actual

---

**Algorithm 1** Decoupling Optimistic Online Mirror Descent

---

1: **Input**: Learning rate $\eta$, initial occupancy measure $g_1 = w_1$
2: Implement the policy reconstructed from $w_1$ on each layer
3: Receive the full information on $c_1$
4: **for** $t = 2, ...T$ **do**
5:     Run Algorithm 2 with costs $c_{t-1}$ and $w_{t-1}$ to compute cumulative cost $\left\{\tilde{c}_{t-1}^l\right\}_{l=0,...,L-1}$
6:     **for** $l = 0, ..., L - 1$ **do**
7:         **for** $x \in \mathcal{X}^l$ **do**
8:             Run Algorithm 4 with $\tilde{c}_{t-1}^l(x)$, $g_{t-1}^l(x)$ and $\eta$ to compute $g_t^l(x)$
9:         **end for**
10:     **end for**
11:     **for** $l = 0, ..., L - 1$ **do**
12:         Receive the realized state $x_t^l$
13:         Receive the prediction from layer $l$ to layer $L$, $M_t^l$
14:         Run Algorithm 3 with $M_t^l$, $\left\{g_t^k(x)\right\}_{k=l,...,L-1, x \in \mathcal{X}^k}$ to compute cumulative prediction $\tilde{M}_t^l$
15:         **for** $x \in \mathcal{X}^l$ **do**
16:             Run Algorithm 4 with $\tilde{M}_t^l(x)$, $g_t^l(x)$ and $\eta$ to compute $w_t^l(x)$
17:         **end for**
18:         Implement the policy reconstructed from $w_t^l(x_t^l)$
19:     **end for**
20:     Receive the full information on $c_t$
21: **end for**

---

or predicted), while maintaining proximity to the previous occupancy measure to ensure robustness. Specifically, we choose $R$ as the unnormalized negative entropy regularizer — for occupancy measure $g$ defined on space $\tilde{\mathcal{U}}$, $R(g) = \sum_{u \in \tilde{\mathcal{U}}} g(u) \log g(u) - \sum_{u \in \tilde{\mathcal{U}}} g(u)$. Under this selection of Legendre, we have $D_R(w, g) = \sum_{u \in \tilde{\mathcal{U}}} w(u) \log \frac{w(u)}{g(u)} - \sum_{u \in \tilde{\mathcal{U}}}[w(u) - g(u)]$, which corresponds to the unnormalized K-L divergence between $w$ and $g$.

## 5   Regret Analysis

In this section, we analyze the regret bound of the DOOMD algorithm, focusing on how each sub-algorithm contributes to the overall performance. We start with the case where all sub-algorithms utilize time-invariant learning rates throughout the $T$ episodes.

### 5.1   Fixed learning rate

As previously demonstrated, $w_t^l(x)$ is a probability distribution condition on reaching state $x$. Therefore, the probability $p_t(x^l, a^l)$ of executing state-action pair $(x^l, a^l)$ at layer $l \geq 1$ in episode $t$ is:

$$p_t(x^l, a^l) = \left[\sum_{x \in \mathcal{A}^{-1}(x^l)} p_t(x, x^l)\right] w_t^l(x^l, a^l). \tag{5}$$

As $p_t$ forms a valid occupancy measure, i.e. $\sum_{u \in \mathcal{U}^l} p_t(u) = 1$ holds for all layer $l$, the algorithm's regret can be expressed as $\mathbf{R}_T = \sum_{t=1}^T \langle c_t, p_t - p^* \rangle$, where $p^* \in \arg\min_{p \in K} \sum_{t=1}^T \langle c_t, p \rangle$. By restricting the region from $K$ to $K^\delta$, the regret can be bounded by:

$$\mathbf{R}_T \leq \sum_{t=1}^T \langle c_t, p_t - p' \rangle + \left|\sum_{t=1}^T \langle c_t, p' - p^* \rangle\right| \leq \sum_{t=1}^T \langle c_t, p_t - p' \rangle + (L-1)\delta T, \tag{6}$$

where $p' \in \arg\min_{p \in K^\delta} \langle c_t, p \rangle$.

**Regret decomposition.** As before, setting $\delta$ sufficiently small controls the second term in Equation (6). Therefore, we primarily focus on bounding the first term, which can be equivalently decomposed to each sub-algorithm, as detailed in the following proposition:

**Proposition 5.1** *For all $p \in K^\delta$, we have:*

$$\sum_{t=1}^{T} \langle c_t, p_t - p \rangle = \sum_{l=0}^{L-1} \sum_{x \in \mathcal{U}^l} \left( \sum_{a \in \mathcal{A}(x)} p(x,a) \right) \sum_{t=1}^{T} \langle \tilde{c}_t^l(x), w_t^l(x) - w(x) \rangle \tag{7}$$

*where $\tilde{c}_t$ is the cumulative costs computed by Algorithm 2, and $w(x,a) = \frac{p(x,a)}{\sum_{a \in \mathcal{A}(x)} p(x,a)}$.*

As each component $\sum_{t=1}^{T} \langle \tilde{c}_t^l(x), w_t^l(x) - w(x) \rangle$ in Proposition 5.1 corresponds to the regret of a sub-algorithm, a critical insight here is that the total regret is bounded if each sub-algorithm performs effectively.

**Prediction accuracy.** As each sub-algorithm is based on the OOMD algorithm, the accuracy of prediction significantly impacts the algorithm's performance. Proposition 5.2 quantifies the precision of cumulative predictions on each layer. Intuitively, besides accumulating errors through layers, the prediction has to make extra sacrifices to handle currently unknown occupancy measures.

**Proposition 5.2** *If the prediction error received on layer $l$ ($0 \leq l \leq L-1$) is bounded by $\epsilon^l$, the prediction error of the cumulative cost is upper bounded by:*

$$\|\tilde{M}_t^l(x) - \tilde{c}_t^l(x)\|_\infty \leq (L-l)\epsilon^l + 2\eta \sum_{m=1}^{L-l-1} m^2 \quad \forall x \in \mathcal{X}^l. \tag{8}$$

For simplicity, denote $Z_l = \sum_{m=1}^{L-l-1} m^2$. Despite this additional error term that may hamper the prediction accuracy, it is already the best prediction we can make given the uncertainty of future decisions. Fortunately, as we are dealing with long horizon $T$, the learning rate $\eta$ is typically very small, such as $\eta = \max_{l=0,\dots,L-1} \sqrt{\frac{\log |\mathcal{U}^l||\mathcal{U}^{l+1}|}{T}}$ in Zimin & Neu (2013). Therefore, the additional term is in the order of $O\left(\frac{L^3}{\sqrt{T}}\right)$.

**Algorithm performance.** The following lemma bridges the gap between prediction accuracy and the sub-algorithm's performance, affirming that tighter control over prediction errors directly contributes to minimizing regret. We skip the proof as it can be easily proved using Proposition 5.2 and Lemma 3 in Rakhlin & Sridharan (2013).

**Lemma 5.3 (Sub-algorithm's regret bound)** *For any $l = 0, \dots, L-1$ and state $x \in \mathcal{X}^l$, we have:*

$$\sum_{t=1}^{T} \langle \tilde{c}_t^l(x), w_t^l(x) - w(x) \rangle \leq \frac{\eta}{2} \left[ (L-l)\epsilon^l + 2\eta Z^l \right]^2 T + \frac{\ln |\mathcal{X}^{l+1}|}{\eta}. \tag{9}$$

If the prediction error bound $\epsilon^l$ is explicitly known for every layer, an optimal learning rate can be selected, resulting in the following regret bound. To see why this achieves a sublinear regret bound, as shown in Zimin & Neu (2013); Dick et al. (2014), the parameter $\delta$ can be set to a sufficiently small value, such as $\delta = \frac{1}{\sqrt{T}}$. This results in a regret bound of the order $O(\sqrt{T})$, guaranteeing the algorithm's performance in the worst-case scenario.

**Theorem 5.4** *The algorithm with $\eta = \sqrt{\frac{2 \sum_{l=0}^{L-1} \ln |\mathcal{X}^{l+1}|}{T \sum_{l=0}^{L-1} [(L-l)\epsilon^l]^2}}$ obtains the following regret bound:*

$$\boldsymbol{R}_T \leq O \left( \sqrt{2 \left( \sum_{l=0}^{L-1} \ln |\mathcal{X}^{l+1}| \right) \left( \sum_{l=0}^{L-1} [(L-l)\epsilon^l]^2 \right) T} + \delta(L-1)T \right). \tag{10}$$

**Flexible learning rates.** To further enhance flexibility, each sub-algorithm on different layers can employ different learning rates. While this variation does not affect the regret decomposition in Proposition 5.1, it influences the accumulation of prediction errors in Proposition 5.2. Denote the learning rate of the sub-algorithm on state $x \in \mathcal{X}^l$ as $\eta^l$. The prediction error bounds follow a similar structure, which is detailed in the following proposition. The proof is omitted as it follows a similar process used in Proposition 5.2. It is evident that if the same learning rate is utilized on each sub-algorithm (i.e., $\eta^l = \eta$ for all $l$), the result reduces to Proposition 5.2.

**Proposition 5.5** *If the prediction error received on layer $l$ ($0 \le l \le L - 1$) is bounded by $\epsilon^l$, the prediction error of the cumulative cost is upper bounded by:*

$$\|\tilde{M}_t^l(x) - \tilde{c}_t^l(x)\|_\infty \le (L - l)\epsilon^l + 2 \sum_{m=l+1}^{L-1} \eta^m (L - m)^2 \quad \forall x \in \mathcal{X}^l. \tag{11}$$

With different learning rates per layer, each sub-algorithm can select its own optimal learning rate, resulting in the following regret bound. Intuitively, utilizing different learning rates across layers introduces more flexibility, enabling the algorithm to perform better. Proposition 5.7 confirms this improvement.

**Theorem 5.6** *The algorithm with $\eta^l = \frac{1}{(L-l)\epsilon^l} \sqrt{\frac{2 \ln |\mathcal{X}^{l+1}|}{T}}$ at layer $l$ obtains the following regret bound:*

$$\boldsymbol{R}_T \le O \left( \sum_{l=0}^{L-1} (L - l)\epsilon^l \sqrt{2 \ln |\mathcal{X}^{l+1}|T} + \delta(L - 1)T \right). \tag{12}$$

**Proposition 5.7** *The algorithm with flexible learning rates (Theorem 5.6) has a lower regret bound compared to one with a uniform learning rate (Theorem 5.4).*

## 5.2 Dynamically updated learning rates

**The doubling trick.** It is worth pointing out that in many realistic cases, the prediction accuracy $\epsilon^l$ may not be explicitly known, making it challenging to determine the optimal learning rate. To address this issue, we employ the doubling trick (Rakhlin & Sridharan, 2013), a common technique in online learning algorithms. It offers a dynamic approach to adjust the learning rate, further improving the adaptability of the algorithm. Specifically, the doubling trick records the accumulated prediction errors, and when the error exceeds a certain threshold, the learning rate is halved, and the accumulated error is reset.

**Algorithm performance.** Denote the learning rate used by the sub-algorithm at state $x \in \mathcal{X}^l$ as $\eta^l(x)$, which is initialized at $\eta_0^l$ and dynamically updated over time. The DOOMD algorithm with dynamically updated learning rates is detailed in Algorithm 5 in Appendix B. By leveraging the doubling trick, the following theorem establishes the algorithm's regret bound.

**Theorem 5.8** *The algorithm with in initial learning rate $\eta_0^l = \frac{1}{2\sqrt{2}(L-l)} \sqrt{\frac{\ln |\mathcal{X}^{l+1}|}{T}}$ at layer $l$ obtains the following regret bound:*

$$\boldsymbol{R}_T \le O \left( \sum_{l=0}^{L-1} 8\sqrt{2}(L - l)\epsilon^l \sqrt{\ln |\mathcal{X}^{l+1}|T} + \delta(L - 1)T \right). \tag{13}$$

**Comparison.** As before, setting $\delta$ to a fairly small value (e.g. $\frac{1}{\sqrt{T}}$) results in a sublinear regret bound. Compared to the regret bound in Theorem 5.6, the theorem above incurs an additional term due to the lack of knowledge about prediction accuracy. However, the algorithm still achieves a sub-linear regret bound of $O(\sqrt{T})$ even without explicit knowledge of the prediction error.

## 6 Numerical Examples

**Experiment setting.** This section provides an empirical verification of the theoretical results. We consider a routing scenario using the METR-LA dataset, a comprehensive record of loop detector data (Jagadish et al., 2014). We evaluate our algorithm in two types of environments: 1) The naturalistic environment that simulates real-world conditions by directly using instantaneous travel time as the prediction; and 2) The adversarial environment that introduces contaminated predictions to test the robustness of our algorithm. The algorithm's performance is compared against three benchmarks: 1) Static benchmark that represents the static optimal policy in hindsight; 2) Greedy benchmark that greedily chooses the outgoing link corresponding to the best route based solely on

predictions; 3) OOMD algorithm that only utilizes the initial prediction without further updates (Rakhlin & Sridharan, 2013). For space reasons, we defer detailed problem descriptions, algorithm setups, and analysis to Appendix D.

**Experiment results.** The performance of the DOOMD algorithm is depicted in Figure 3. The horizontal axis refers to the time scale, and the vertical axis represents the cost difference between the proposed DOOMD algorithm and the three benchmarks, with lower values showcasing our algorithm's superiority. Figure 3(a)-(c) corresponding to the naturalistic environment under different fixed learning rates. The results indicate that with an appropriate learning rate, the DOOMD algorithm outperforms the benchmarks. However, the performance gap is modest due to the reliability of naturalistic predictions. Under the adversarial environment shown in Figure 3(d)-(e), the DOOMD algorithm demonstrates remarkable robustness. For these tests, we fix the learning rate at 5 and vary the attack intensity (described in detail in Appendix D) from 1 to 5. Although increasing attack intensity affects the DOOMD algorithm's performance, its impact is notably milder compared to that of the other benchmarks. Remarkably, even with a moderate attack level, our algorithm substantially outperforms the greedy benchmark.

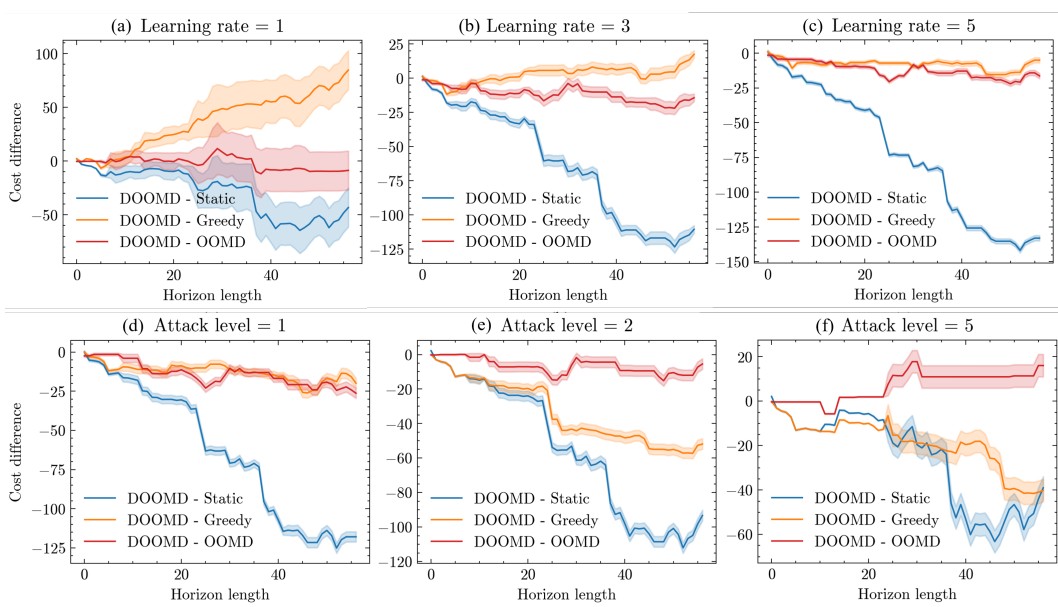

Figure 3: Performance comparison under naturalistic and adversarial environments

## 7 CONCLUSION

In this paper, we have introduced the Decoupled Optimistic Online Mirror Descent (DOOMD) algorithm, a novel online learning approach for episodic MDPs with dynamically updated and improving predictions. The algorithm effectively decomposes decisions across different layers and states, with each state executing a sub-algorithm that accounts for both immediate and long-term effects. We have theoretically analyzed the prediction accuracy and established a sublinear regret bound of the DOOMD algorithm, underscoring the algorithm's robustness in worst-case scenarios.

For future work, an interesting direction is to extend our model to a bandit feedback setting, where the learner only observes the true costs of the selected state-action pairs. This transition poses significant challenges in accurately estimating costs from limited information but could greatly enhance the algorithm's practical applicability. Additionally, analyzing dynamic regret would be valuable to further understand and quantify the algorithm's performance over time.

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

# A    APPENDIX A. NOTATION TABLE

| | Sets |
|---|---|
| $\mathcal{T}$ | Time horizon |
| $S$ | Path set |
| $\mathcal{L}$ | Layers |
| $\mathcal{X}$ | State space |
| $\mathcal{X}^l$ | States on layer $l$ |
| $\mathcal{U}$ | State-action pairs |
| $\mathcal{U}^l$ | State-action pairs on layer $l$ |
| $\mathcal{U}^{l:k}$ | State-action pairs from layer $l$ to $k$ |
| $\mathcal{A}(x)$ | Accessible actions for state $x$ |
| $\mathcal{A}^{-1}(x)$ | States with $x$ as an accessible action |
| $K$ | Definition domain for occupancy measures |
| $K^\delta$ | Restricted definition domain for occupancy measures |
| $\mathcal{U}^l(x)$ | State-action pairs for state $x$ on layer $l$ |
| | **Variables** |
| $x_t^l$ | State at layer $l$ on day $t$ |
| $a_t^l$ | Action at layer $l$ on day $t$ |
| $w_t, g_t$ | Occupancy measures on day $t$ |
| $w_t^l(x), g_t^l(x)$ | Occupancy measures at state $x \in \mathcal{X}^l$ on day $t$ |
| $p_t(x, a)$ | Probability of executing $(x, a)$ on day $t$ |
| | **Paramters** |
| $\epsilon^l$ | Error bound of prediction received on layer $l$ |
| $Z^l$ | A constant term in regret for layer $l$ |
| | **Functions** |
| $p(x'\|x, a)$ | Transition kernel |
| $c_t(x, a)$ | Cost function on day $t$ |
| $M_t^l(x, a)$ | Cost prediction received at layer $l$ on day $t$ |
| $\pi_t^l(a\|x)$ | Policy implemented at layer $l$ on day $t$ |
| $\mathbf{R}_T$ | Total regret |
| $\tilde{c}_t^l(x, a)$ | cumulative cost at layer $l$ on day $t$ |
| $c_t(x)$ | Cost of state-action pairs for state $x$ on day $t$ |
| $M_t^l(x)$ | Cost predictions received on layer $l$ for state-action pairs at state $x$ on day $t$ |
| $\tilde{M}_t^l(x, a)$ | cumulative prediction at layer $l$ on day $t$ |

# B  APPENDIX B. ALGORITHM

---

**Algorithm 2** Computation of cumulative costs

---

1: **Input**: Cost $c \in [0,1]^{|\mathcal{U}^{0:L-1}|}$, occupancy measure $\left\{w^l(x)\right\}_{l=0,\ldots,L-1,x\in\mathcal{X}^l}$
2: **Output**: The cumulative cost $\left\{\tilde{c}^l\right\}_{l=0,\ldots,L-1}$
3: **for** $l = L-1, \ldots, 0$ **do**
4:     **for** $x \in \mathcal{X}^l$ **do**
5:         **for** $a \in \mathcal{A}(x)$ **do**
6:             Compute the cumulative cost:
7:             $$\tilde{c}^l(x,a) = \begin{cases} c(x,a) + \langle w^{l+1}(a), \tilde{c}^{l+1}(a)\rangle & \text{if } l \neq L-1 \\ c(x,a) & \text{otherwise} \end{cases}$$
8:         **end for**
9:     **end for**
10: **end for**

---

---

**Algorithm 3** Compute predictions of the cumulative cost

---

1: **Input**: Prediction $M \in [0,1]^{|\mathcal{U}^{l:L-1}|}$, occupancy measure $\left\{g^k(x)\right\}_{k=l,\ldots,L-1,x\in\mathcal{X}^k}$
2: **Output**: Prediction of the cumulative cost $\tilde{M}^l$
3: **for** $k = L-1, \ldots, l$ **do**
4:     **for** $x \in \mathcal{X}^k$ **do**
5:         **for** $a \in \mathcal{A}(x)$ **do**
6:             Compute the cumulative prediction:
7:             $$\tilde{M}^k(x,a) = \begin{cases} M(x,a) + \langle g^{k+1}(a), \tilde{M}^{k+1}(a)\rangle & \text{if } k \neq L-1 \\ M(x,a) & \text{otherwise} \end{cases}$$
8:         **end for**
9:     **end for**
10: **end for**

---

---

**Algorithm 4** One-step update based on OOMD

---

1: **Input**: Occupancy measure $g$ on some space $\tilde{\mathcal{U}}$, cost $c \in [0,1]^{|\tilde{\mathcal{U}}|}$, learning rate $\eta$
2: **Output**: Occupancy measure $g^+$ on $\tilde{\mathcal{U}}$
3: Compute $g^+ = \arg\min_w \left\{\eta\langle c, w\rangle + D_R(w, g)\right\}$

---

---

**Algorithm 5** Decoupling Optimistic Online Mirror Descent with the doubling trick

---

1: **Input**: Learning rate $\eta^l(x) = \eta_0^l$ for each sub-algorithm on state $x \in \mathcal{X}^l$, initial occupancy measure $g_1 = w_1$, initial prediction error $E_1^l(x) = 0$ for every layer $l$ and $x \in \mathcal{X}^l$
2: Implement the policy reconstructed from $w_1$ and run Algorithm 3 to compute $\tilde{M}_1^l$ on each layer
3: Receive the full information on $c_1$
4: **for** $t = 2, ...T$ **do**
5:     **for** $l = L-1, ..., 0$ **do**
6:         **for** $x \in \mathcal{X}^l$ **do**
7:             **for** $a \in \mathcal{A}(x)$ **do**
8:                 Compute the accumulated cost:
9:
$$\tilde{c}_{t-1}^l(x,a) = \begin{cases} c_{t-1}(x,a) + \langle w_{t-1}^{l+1}(a), \tilde{c}_{t-1}^{l+1}(a)\rangle & \text{if } k \neq L-1 \\ c_{t-1}(x,a) & \text{otherwise} \end{cases}$$
10:             **end for**
11:             Update the accumulated prediction error $E_t^l(x) = E_{t-1}^l(x) + \|\tilde{M}_{t-1}^l(x) - \tilde{c}_{t-1}^l(x)\|_\infty$
12:             **if** $\frac{\eta^l(x)}{2} E_t^l(x) > \frac{1}{\eta^l(x)}$ **then**
13:                 $\eta^l(x) = \eta^l(x)/2$
14:                 $E_t^l(x) = 0$
15:             **end if**
16:             Compute one-step update:
17:                 $g_t^l(x) = \arg\min_w \left\{ \eta^l(x)\langle \tilde{c}_{t-1}^l(x), w\rangle + D_R(w, g_{t-1}^l(x)) \right\}$
18:         **end for**
19:     **end for**
20:     **for** $l = 0, ..., L-1$ **do**
21:         Receive the realized state $x_t^l$
22:         Receive the prediction from layer $l$ to layer $L$, $M_t^l$
23:         Run Algorithm 3 with $M_t^l$, $\{g_t^k(x)\}_{k=l,...,L-1, x \in \mathcal{X}^k}$ to compute cumulative prediction $\tilde{M}_t^l$
24:         **for** $x \in \mathcal{X}^l$ **do**
25:             Compute the second update:
26:             $w_t^l(x) = \arg\min_w \left\{ \eta^l(x)\langle \tilde{M}_t^l, w\rangle + D_R(w, g_t^l(x)) \right\}$
27:         **end for**
28:          Implement the policy reconstructed from $w_t^l(x_t^l)$
29:     **end for**
30:     Receive the full information $c_t$
31: **end for**

---

# C APPENDIX C. STOCHASTIC TRANSITION

So far, we have focused on deterministic transitions to better convey the main ideas. This section extends the analysis to a general stochastic transition function $P$. In this case, if action $a \in \mathcal{A}(x)$ is taken at the current state $x$, the state will transition to $x'$ with probability $P(x'|x,a)$. To maintain the layered structure, for $x \in \mathcal{X}^l, l = 0, ..., L-1$, we require that if $P(x'|x,a) > 0$, it must hold that $x' \in \mathcal{X}^l$. For state-action pair $u = (x,a)$, for simplicity, we sometimes write the transition function as $P(x'|u) = P(x'|x,a)$.

With stochastic transitions, the domain of occupancy measure is redefined as:

$$K = \left\{ w : \sum_{u \in \mathcal{U}^l} w(u) = 1, \sum_{a \in \mathcal{A}(x)} w(x,a) = \sum_{u \in \mathcal{U}^l} w(u)P(x'|u), \forall l \in [L-1], x \in \mathcal{X}^{l+1} \right\}, \quad (14)$$

While the primary algorithm (Algorithm 1) remains unchanged, sub-algorithms for constructing cumulative costs and predictions must be adjusted. These adjustments are detailed in Algorithm 6 and Algorithm 7, respectively.

---

**Algorithm 6** Computation of cumulative costs with stochastic transitions

---

1: **Input**: Cost $c \in [0,1]^{|\mathcal{U}^{0:L-1}|}$, occupancy measure $\left\{w^l(x)\right\}_{l=0,\ldots,L-1,x\in\mathcal{X}^l}$

2: **Output**: The cumulative cost $\left\{\tilde{c}^l\right\}_{l=0,\ldots,L-1}$

3: **for** $l = L-1,\ldots,0$ **do**

4:     **for** $x \in \mathcal{X}^l$ **do**

5:         **for** $a \in \mathcal{A}(x)$ **do**

6:             Compute the cumulative cost:

7:
$$
\tilde{c}^l(x,a) = \begin{cases} c(x,a) + \displaystyle\sum_{s\in\mathcal{X}^{l+1}} P(s|x,a)\langle w^{l+1}(s), \tilde{c}^{l+1}(s)\rangle & \text{if } l \neq L-1 \\ c(x,a) & \text{otherwise} \end{cases}
$$

8:         **end for**

9:     **end for**

10: **end for**

---

**Algorithm 7** Compute predictions of the cumulative cost with stochastic transitions

---

1: **Input**: Prediction $M \in [0,1]^{|\mathcal{U}^{l:L-1}|}$, occupancy measure $\left\{g^k(x)\right\}_{k=l,\ldots,L-1,x\in\mathcal{X}^k}$

2: **Output**: Prediction of the cumulative cost $\tilde{M}^l$

3: **for** $k = L-1,\ldots,l$ **do**

4:     **for** $x \in \mathcal{X}^k$ **do**

5:         **for** $a \in \mathcal{A}(x)$ **do**

6:             Compute the cumulative prediction:

7:
$$
\tilde{M}^k(x,a) = \begin{cases} M(x,a) + \displaystyle\sum_{s\in\mathcal{X}^{l+1}} P(s|x,a)\langle g^{k+1}(s), \tilde{M}^{k+1}(s)\rangle & \text{if } k \neq L-1 \\ M(x,a) & \text{otherwise} \end{cases}
$$

8:         **end for**

9:     **end for**

10: **end for**

---

Compared with the deterministic transition case, the primary adjustment is in Line 7, where all possible transitions for each state-action pair $(x,a)$ are now considered. In the deterministic transition case (i.e., $P(s|x,a) = 1$ if and only if $s = a$), these algorithms reduce to their previous formulations.

Building on Equation (6), we can decompose the first component in a similar manner. The result is summarized in the following proposition:

**Proposition C.1** *For all $p \in K^\delta$, we have:*

$$
\sum_{t=1}^{T}\langle c_t, p_t - p\rangle = \sum_{l=0}^{L-1}\sum_{x\in\mathcal{U}^l}\left(\sum_{a\in\mathcal{A}(x)}p(x,a)\right)\sum_{t=1}^{T}\langle\tilde{c}_t^l(x), w_t^l(x) - w(x)\rangle \tag{15}
$$

*where $\tilde{c}_t$ is the cumulative costs computed by Algorithm 6, and $w(x,a) = \frac{p(x,a)}{\sum_{a\in\mathcal{A}(x)}p(x,a)}$.*

Note that the decomposition does not have fundamental changes despite the new formulation of cumulative costs by Algorithm 6. The following proposition bounds the error of the cumulative predictions constructed by Algorithm 7.

**Proposition C.2** *If the prediction error received on layer $l$ ($0 \leq l \leq L-1$) is bounded by $\epsilon^l$, the prediction error of the cumulative cost is upper bounded by:*

$$
\|\tilde{M}_t^l(x) - \tilde{c}_t^l(x)\|_\infty \leq (L-l)\epsilon^l + 2\eta\sum_{m=1}^{L-l-1}m^2 \quad \forall x \in \mathcal{X}^l. \tag{16}
$$

Note that each sub-algorithm in Algorithm 1 maintains control over cumulative costs despite the reformulated computations. Therefore, transitioning from deterministic to stochastic transitions does not fundamentally alter the regret analysis. We skip the proof because it is identical to Theorem 5.4.

**Theorem C.3** *The algorithm with* $\eta = \sqrt{\frac{2\sum_{l=0}^{L-1}\ln|\mathcal{X}^{l+1}|}{T\sum_{l=0}^{L-1}[(L-l)\epsilon^l]^2}}$ *obtains the following regret bound:*

$$\boldsymbol{R}_T \leq O\left(\sqrt{2\left(\sum_{l=0}^{L-1}\ln|\mathcal{X}^{l+1}|\right)\left(\sum_{l=0}^{L-1}[(L-l)\epsilon^l]^2\right)T} + \delta(L-1)T\right). \tag{17}$$

For dynamically updated learning rates, the regret bounds can similarly be extended as before, which is omitted in this paper.

## D APPENDIX D. NUMERICAL EXAMPLES

### D.1 EXPERIMENT SETTING

In this experiment, we utilize the METR-LA dataset, a comprehensive record of loop detector data in the highway of Los Angeles County (Jagadish et al., 2014) to simulate real-world conditions. We utilize traffic speed data recorded every 5 minutes by 13 selected loop detectors, labeled $A$ to $M$. These detectors, viewed as nodes, are interconnected in a simplified network consisting of 14 links, as shown in Figure 4. The speed recorded at the start of each link serves as the constant travel speed on the entire level. For instance, the speed recorded by detector $B$ at 8:10 am dictates the travel speed on link 2 from 8:10 to 8:15 am. Additionally, to accommodate nodes with multiple exiting links, speeds from five auxiliary detectors (labeled $v$ to $z$) are used to determine the speed on each distinct outgoing link. Specifically, the speed recorded at node $w$, $x$, $y$, $v$, and $z$ is used for link 1, 3, 8, 10, and 14, respectively.

By integrating the location data of each loop detector (Li et al., 2017), we calculate the distance between each node, thereby deriving the link travel time for every timestep. The dataset spans 4 months from March 1st, 2012 to June 30th, 2012. After preprocessing, there are 57 days with valid data, establishing our experiment's temporal scope.

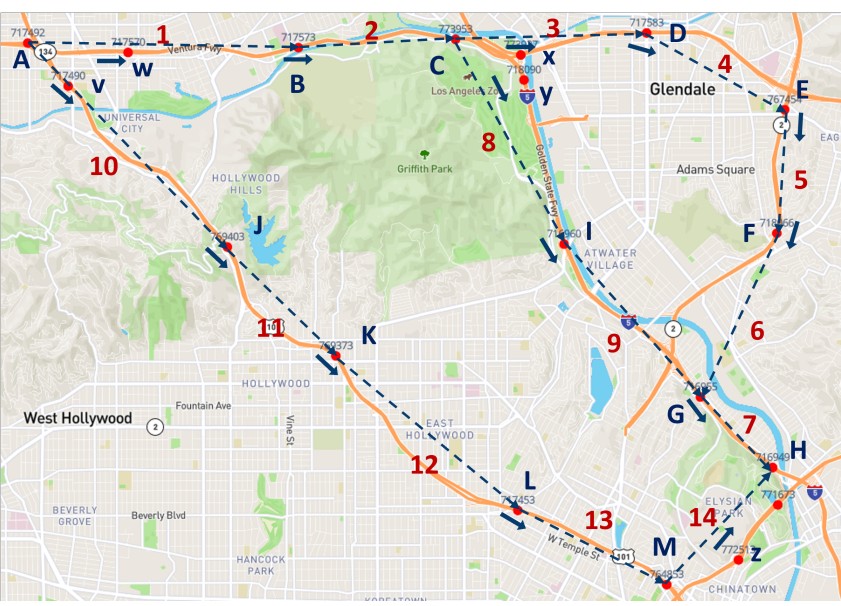

Figure 4: A simplified network in Los Angeles

This experiment focuses on a virtual vehicle routing from node $A$ to $H$ at 8:00 am daily, navigating through three potential paths. Following our modeling approach, the routing problem is simplified to a layered structure in Figure 5. Here, node $L$ is added to complete the layered structure. Specifically, state-action pairs $C - H$ and $C - H'$ represent paths $C - D - E - F - G - H$ or $C - I - G - H$,

respectively. The cost of each state-action pair refers to the corresponding travel time, which can be calculated recursively from the travel times on the respective links.

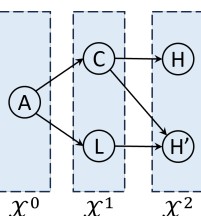

Figure 5: Equivalent layered structure for the path-planning scenario

## D.2 SCENARIOS AND BENCHMARKS

**Experiment scenarios.** This experiment evaluates the proposed algorithm under two distinct scenarios:

- Naturalistic environment: The instantaneous travel time, which displays the current travel time on each link, is directly used as the prediction.

- Adversarial environment: Incorporates a simple attack strategy designed to contaminate the predictions and make the environment more adversarial. The predictions between the two best actions at each decision point are skewed: The attack adds $\beta$ to the prediction of the best-anticipated action, and minus $\beta$ to the second-best one, where $\beta$ represents the attack level.

**Benchmarks.** For the greedy benchmark, at every decision point, the algorithm calculates the predicted travel time on all the potential route choices and selects the first link in the optimal predicted route. For the OOMD benchmark, it implements a single OOMD algorithm (Rakhlin & Sridharan, 2013), which can be seen as a pre-trip routing strategy that only utilizes the initial prediction at the origin.

## D.3 EXPERIMENT RESULTS

**Naturalistic environment.** In Figure 3(a)-(c), a fixed learning rate is applied across all sub-algorithms. The DOOMD and OOMD algorithms are executed five times for each experiment to eliminate the influence of the stochastic policy, with each solid curve representing the mean cost difference and the shaded region indicating the standard deviation.

The blue and orange curves highlight a preference for higher learning rates, which can be attributed to the reliable nature of naturalistic predictions. While these predictions may not always precisely match the true costs, they reliably indicate the relative magnitudes, generally guiding the selection toward the optimal decisions. A higher learning rate enhances the algorithm's dependency on these predictions, thus improving performance. Notably, at a learning rate of 5, DOOMD outperforms the greedy benchmark, indicating its capability to handle naturalistic prediction errors. Further fine-tuning, such as adjusting learning rates for different layers, might enhance performance, but it is beyond this paper's scope. Figure 3 also reveals that our algorithm greatly outperforms the static benchmark. Note that it does not mean the sublinear bound we obtained in Theorem 5.8 is meaningless as the naturalistic predictions do not represent the worst-case scenario. Additionally, the real-time information contained in the updated predictions benefits the DOOMD algorithm, leading to superior performance compared to the OOMD benchmark, as shown by the red curves.

**Adversarial environment** For these tests, we fix the learning rate at 5 while varying attack level $\beta$ from 1 to 5. Although increasing attack intensity affects the algorithm's performance, its impact is notably milder compared to that on the greedy benchmark. Remarkably, even with a moderate attack level ($\beta = 2$), our algorithm substantially outperforms the greedy benchmark, highlighting its robustness in adversarial settings. Another interesting observation emerges at the highest attack level, where the OOMD benchmark momentarily outperforms DOOMD. It is because under heavy

perturbations, leveraging updated prediction on Node $C$ is counterproductive. This suggests that in highly compromised environments, a strategy that reduces reliance on incoming predictions could be more effective, suggesting a potential shift in algorithm design when facing severely adversarial environments.

## APPENDIX E. PROOFS

### PROOF FOR PROPOSITION 5.1

We prove this proposition by induction on $L$. Let us start with an MDP with 2 layers, i.e. $L = 1$. In this special case, $\tilde{c}_t^0 = c_t$ for all $t = 1, ..., T$, and $w_t^l(x_0) = p_t(x_0)$. Thus, the equivalency immediately holds.

Assume that the proposition holds for all MDPs with $L = K$ ($K \geq 1$). Consider any MDP with $L = K + 1$, the expected cost difference at state $x \in \mathcal{X}^K$ on day $t$ can be expressed as:

$$\sum_{a^K \in \mathcal{A}(x)} c_t(x, a^K) \left[ p_t(x, a^K) - p(x, a^K) \right]$$

$$= \sum_{a^K \in \mathcal{A}(x)} c_t(x, a^K) \left[ p_t(x, a^K) - w_t^K(x, a^K) \sum_{a \in \mathcal{A}(x)} p(x, a) \right] \qquad (18)$$

$$+ \sum_{a^K \in \mathcal{A}(x)} c_t(x, a^K) \left[ w_t^K(x, a^K) \sum_{a \in \mathcal{A}(x)} p(x, a) - p(x, a^K) \right].$$

The second component is equivalent to:

$$\left( \sum_{a \in \mathcal{A}(x)} p(x, a) \right) \sum_{a^K \in \mathcal{A}(x)} c_t(x, a^K) \left[ w_t^K(x, a^K) - \frac{p(x, a^K)}{\sum_{a \in \mathcal{A}(x)} p(x, a)} \right]$$

$$= \left( \sum_{a \in \mathcal{A}(x)} p(x, a) \right) \langle c_t(x), w_t^K(x) - w(x) \rangle, \qquad (19)$$

which reflects the contribution of the sub-algorithm on state $x$. Note that $w(x)$ is always well-defined as $p \in K^\delta$. Meanwhile, by leveraging definition and the flow conservation of $p$, the first component is equivalent to:

$$\sum_{a^K \in \mathcal{A}(x)} c_t(x, a^K) \left[ \left( \sum_{s \in \mathcal{A}^{-1}(x)} p_t(s, x) \right) w_t^K(x, a^K) - w_t^K(x, a^K) \left( \sum_{s \in \mathcal{A}^{-1}(x)} p(s, x) \right) \right]$$

$$= \left[ \sum_{s \in \mathcal{A}^{-1}(x)} (p_t(s, x) - p(s, x)) \right] \langle c_t(x), w_t^K(x) \rangle, \qquad (20)$$

which is equivalently credited to sub-algorithms on earlier layers. In this sense, every state-action pair $(s, x) \in \mathcal{U}^{K-1}$ shares $(p_t(s, x) - p(s, x)) \langle c_t(x), w_t^K(x) \rangle$ from layer $K$. Combining with its immediate contribution, the total cost difference on this pair is $(p_t(s, x) - p(s, x)) (c_t(s, x) + \langle c_t(x), w_t^K(x) \rangle)$, which exactly matches the cumulative cost $\tilde{c}_t^{K-1}(s, x)$ computed by Algorithm 2.

Let us treat layer $K$ as the final layer by neglecting layer $K + 1$, and take $\tilde{c}_t^{K-1}$ as the actual cost on $\mathcal{U}^{K-1}$, which does not influence any further previous layer. Due to the induction assumption, the cost difference on the remaining $K - 1$ layers can be expressed as:

$$\sum_{l=0}^{K-1} \sum_{x \in \mathcal{U}^l} \left( \sum_{a \in \mathcal{A}(x)} p(x, a) \right) \sum_{t=1}^{T} \langle \tilde{c}_t^l(x), w_t^l(x) - w(x) \rangle. \qquad (21)$$

Moreover, as $c_t(x) = \tilde{c}_t^K(x)$ holds for all $x \in \mathcal{X}^K$ by definition, adding Equation (19) yields the total cost difference as

$$\sum_{l=0}^{K} \sum_{x \in \mathcal{U}^l} \left( \sum_{a \in \mathcal{A}(x)} p(x,a) \right) \sum_{t=1}^{T} \langle \tilde{c}_t^l(x), w_t^l(x) - w(x) \rangle. \tag{22}$$

Thus, the proposition also holds for MDPs with $L = K + 1$. By induction, the proposition is proved. $\square$

PROOF FOR PROPOSITION 5.2

To compute $\tilde{M}_t^l$, Algorithm 3 recursively calculates the cumulative predictions $\tilde{M}_t^k$ from $k = L - 1$ to $k = l$. Similarly, Algorithm 2 recursively computes $\tilde{c}_t^k$ from $k = L - 1$ to $k = l$. Let us prove the following result by induction:

$$\|\tilde{M}_t^k(x) - \tilde{c}_t^k(x)\|_\infty \le (L-k)\epsilon^l + 2\eta \sum_{m=1}^{L-k-1} m^2, \quad l \le k \le L-1, \forall x \in \mathcal{X}^k. \tag{23}$$

First, in the case when $k = L - 1$, we have $\tilde{M}_t^{L-1}(x,a) = M_t^l(x,a)$ and $\tilde{c}_t^{L-1}(x,a) = c_t(x,a)$ for all $(x,a) \in \mathcal{U}^{L-1}$ by definition, thus the proposition immediately holds. Assume the proposition holds when $k = K + 1$ ($l \le K \le L - 2$), that is $|\tilde{M}_t^{K+1}(x,a) - \tilde{c}_t^{K+1}(x,a)| \le (L-K-1)\epsilon^l + 2\eta \sum_{m=1}^{L-K-2} m^2$ holds for all $(x,a) \in \mathcal{U}^{K+1}$.

Then, for $x \in \mathcal{X}^K$, recall that

$$\tilde{c}_t^K(x,a) = c_t(x,a) + \langle \tilde{c}_t^{K+1}(a), w_t^{K+1}(a) \rangle, \tag{24}$$

$$\tilde{M}_t^K(x,a) = M_t^l(x,a) + \langle \tilde{M}_t^{K+1}(a), g_t^{K+1}(a) \rangle, \tag{25}$$

which yields the following results for $x \in \mathcal{X}^K$ and $s \in \mathcal{A}(x)$:

$$
\begin{aligned}
&|\tilde{M}_t^K(x,s) - \tilde{c}_t^K(x,s)| \\
&= |M_t^l(x,s) + \langle \tilde{M}_t^{K+1}(s), g_t^{K+1}(s) \rangle - c_t(x,s) - \langle \tilde{c}_t^{K+1}(s), w_t^{K+1}(s) \rangle| \\
&\le \epsilon^l + \sum_{a \in \mathcal{A}(s)} w_t^{K+1}(s,a) \left| \tilde{M}_t^{K+1}(s,a) - \tilde{c}_t^{K+1}(s,a) \right| + \\
&\quad + \sum_{a \in \mathcal{A}(s)} \tilde{M}_t^{K+1}(s,a) \left| g_t^{K+1}(s,a) - w_t^{K+1}(s,a) \right| \\
&\le (L-k)\epsilon^l + 2\eta \sum_{m=1}^{L-K-2} m^2 + \|\tilde{M}_t^{K+1}(s)\|_\infty \|g_t^{K+1}(s) - w_t^{K+1}(s)\|_1
\end{aligned} \tag{26}
$$

where the last inequality is due to Hölder's inequality. Note that $w_t^{K+1}(s)$ minimizes $\eta\langle \tilde{M}_t^{K+1}(s), w \rangle + D_R(w, g_t^{K+1}(s))$, hence

$$\eta\langle \tilde{M}_t^{K+1}(s), w_t^{K+1}(s) \rangle + D_R(w_t^{K+1}(s), g_t^{k+1}(s)) \le \eta\langle \tilde{M}_t^{K+1}(s), g_t^{K+1}(s) \rangle, \tag{27}$$

which leads to:

$$\eta\langle \tilde{M}_t^{K+1}(s), g_t^{K+1}(s) - w_t^{K+1}(s) \rangle \ge D_R(w_t^{K+1}(s), g_t^{K+1}(s)). \tag{28}$$

Leveraging Hölder's inequality and the strong convexity of $R$, we further have:

$$2\eta\|\tilde{M}_t^{K+1}(s)\|_\infty \ge \|g_t^{K+1}(s) - w_t^{K+1}(s)\|_1. \tag{29}$$

Hence, the prediction error is:

$$
\begin{aligned}
|\tilde{M}_t^K(x,s) - \tilde{c}_t^K(x,s)| &\le (L-k)\epsilon^l + 2\eta \sum_{m=1}^{L-K-2} m^2 + 2\eta\|\tilde{M}_t^{K+1}(s)\|_\infty^2 \\
&\le (L-k)\epsilon^l + 2\eta \sum_{m=1}^{L-K-2} m^2 + 2\eta(L-K-1)^2.
\end{aligned} \tag{30}
$$

where the last inequality is due to the upper bound of the cumulative predictions. Therefore, the proposition holds for layer $K$. By induction, the proposition is proved. $\square$

PROOF FOR THEOREM 5.4

For simplicity, denote $\Xi = \frac{2\sum_{l=0}^{L-1}\ln|\mathcal{X}^{l+1}|}{\sum_{l=0}^{L-1}[(L-l)\epsilon^l]^2}$, thus $\eta = \sqrt{\frac{\Xi}{T}}$. According to Equation (6) and Proposition 5.1, the algorithm's regret can be written as:

$$
\begin{aligned}
\mathbf{R}_T \leq & \eta \sum_{l=0}^{L-1} \frac{1}{2}\left[(L-l)\epsilon^l\right]^2 T + \frac{1}{\eta}\sum_{l=0}^{L-1}\ln|\mathcal{X}^{l+1}| \\
& + \sum_{l=0}^{L-1}\left[2\eta^3(Z^l)^2 + 2\eta^2(L-l)\epsilon^l Z^l\right]T + \delta(L-1)T \\
= & \sqrt{2\left(\sum_{l=1}^{L-1}\ln|\mathcal{X}^{l+1}|\right)\left(\sum_{l=0}^{L-1}[(L-l)\epsilon^l]^2\right)T} \\
& + \sum_{l=0}^{L-1}\left[2\Xi\sqrt{\Xi}(Z^l)^2\frac{1}{\sqrt{T}} + 2\Xi(L-l)\epsilon^l Z^l\right] + \delta(L-1)T
\end{aligned}
\tag{31}
$$

As the middle term appears in the order of $O(1)$, which does not affect the order of the regret bound, the theorem is proved. $\square$

PROOF OF THEOREM 5.6

When different learning rates are employed across layers, Lemma 5.3 should be revised accordingly:

$$
\sum_{t=1}^{T}\langle\tilde{c}_t^l(x), w_t^l(x) - w(x)\rangle \leq \frac{\eta^l}{2}\sum_{t=1}^{T}\|\tilde{M}_t^l - \tilde{c}_t^l\|_\infty^2 + \frac{\ln|\mathcal{X}^{l+1}|}{\eta^l}.
\tag{32}
$$

For simplicity, denote $\Xi_l = 2\sum_{m=l+1}^{L-1}\eta^m(L-m)^2 = 2\sum_{m=l+1}^{L-1}\frac{L-m}{\epsilon^m}\sqrt{\frac{2\ln|\mathcal{X}^{m+1}|}{T}}$, which is in the order of $O\left(\sqrt{\frac{1}{T}}\right)$. According to Equation (6) and Proposition 5.1, the algorithm's regret can be written as:

$$
\mathbf{R}_T \leq \sum_{l=0}^{L-1}\left\{\frac{\eta^l}{2}\left[(L-l)\epsilon^l\right]^2 T + \frac{\ln|\mathcal{X}^{l+1}|}{\eta^l}\right\} + \sum_{l=0}^{L-1}\left[\frac{\eta^l}{2}\Xi_l^2 + \eta^l(L-l)\epsilon^l\Xi_l\right]T,
\tag{33}
$$

where the last term is in the order of $O(1)$. Substituting the value of $\eta^l$ yields the regret bound in the theorem. $\square$

PROOF FOR PROPOSITION 5.7

The proposition can be proved by applying the Cauchy-Schiwtz inequality on Equation (10) and Equation (12). $\square$

PROOF FOR THEOREM 5.8

As the doubling trick only decreases or maintains the learning rate, the prediction error of the cumulative cost for any day $t$ can be upper bounded by:

$$
\|\tilde{M}_t^l(x) - \tilde{c}_t^l(x)\|_\infty \leq (L-l)\epsilon^l + 2\sum_{m=l+1}^{L-1}\eta_0^m(L-m)^2 \quad \forall x \in \mathcal{X}^l.
\tag{34}
$$

For each sub-algorithm on each layer $l$, as shown by Lemma 12 in Rakhlin & Sridharan (2013), if its learning rate is never updated in the process, the regret is bounded by:

$$
\begin{aligned}
\sum_{t=1}^{T}\langle\tilde{c}_t^l(x), w_t^l(x) - w(x)\rangle & \leq \frac{4\ln|\mathcal{X}^{l+1}|}{\eta_0^l} \\
& \leq 8\sqrt{2}(L-l)\sqrt{\ln|\mathcal{X}^{l+1}|T};
\end{aligned}
\tag{35}
$$

otherwise, the regret is upper bounded by:

$$\sum_{t=1}^{T} \langle \tilde{c}_t^l(x), w_t^l(x) - w(x) \rangle \leq 8\sqrt{2} \sqrt{E\left[\sum_{t=1}^{T} \|\tilde{M}_t^l - \tilde{c}_t^l\|_\infty^2\right] \ln|\mathcal{X}^{l+1}|}$$

$$\leq 8\sqrt{2\ln|\mathcal{X}^{l+1}|T}\left[(L-l)\epsilon^l + 2\sum_{m=l+1}^{L-1} \eta_0^m (L-m)^2\right]. \tag{36}$$

As the cost is normalized to $[0, 1]$, the prediction error naturally should satisfy $\epsilon^l \leq 1$. Therefore, combining the two cases yields:

$$\sum_{t=1}^{T} \langle \tilde{c}_t^l(x), w_t^l(x) - w(x) \rangle$$

$$\leq 8\sqrt{2}(L-l)\epsilon^l\sqrt{\ln|\mathcal{X}^{l+1}|T} + 16\sqrt{2\ln|\mathcal{X}^{l+1}|T}\left[\sum_{m=l+1}^{L-1} \eta_0^m (L-m)^2\right]. \tag{37}$$

According to Equation (6) and Proposition 5.1, the algorithm's regret can be written as:

$$\mathbf{R}_T \leq \sum_{l=1}^{L-1}\left\{8\sqrt{2}(L-l)\epsilon^l\sqrt{\ln|\mathcal{X}^{l+1}|T} + 16\sqrt{2\ln|\mathcal{X}^{l+1}|T}\left[\sum_{m=l+1}^{L-1} \eta_0^m (L-m)^2\right]\right\} + \delta(L-1)T. \tag{38}$$

Omitting the middle term, which appears in the order of $O(1)$ and does not influence the overall order of the regret bound, we prove the theorem. $\square$

PROOF FOR PROPOSITION C.1

We prove this proposition by induction on $L$. Let us start with an MDP with 2 layers, i.e. $L = 1$. In this special case, $\tilde{c}_t^0 = c_t$ for all $t = 1, ..., T$, and $w_t^l(x_0) = p_t(x_0)$. Thus, the equivalency immediately holds.

Assume that the proposition holds for all MDPs with $L = K$ ($K \geq 1$). Consider any MDP with $L = K + 1$, the expected cost difference at state $x \in \mathcal{X}^K$ on day $t$ can be splitter into the same two components as in Equation (18), and the latter can be rewritten as Equation (19).

Due to the flow conservation $\sum_{u \in \mathcal{U}^{K-1}} p(u)P(x|u) = \sum_{a \in \mathcal{A}(x)} p(x, a)$, the former part can be rewritten as:

$$\sum_{a^K \in \mathcal{A}(x)} c_t(x, a^K)\left[\left(\sum_{u \in \mathcal{U}^{K-1}} p_t(u)P(x|u)\right)w_t^K(x, a^K) - w_t^K(x, a^K)\left(\sum_{u \in \mathcal{U}^{K-1}} p(u)P(x|u)\right)\right]$$

$$= \left[\sum_{u \in \mathcal{U}^{K-1}} P(x|u)(p_t(u) - p(u))\right]\langle c_t(x), w_t^K(x)\rangle, \tag{39}$$

In this sense, every state-action pair $(s, a) \in \mathcal{U}^{K-1}$ shares

$$P(x|s, a)\langle c_t(x), w_t^K(x)\rangle(p_t(s, a) - p(s, a)). \tag{40}$$

from state $x$ in layer $K$. Therefore, combining with all other states in layer $K$ and its immediate contribution, the total cost difference on this pair is:

$$\left(c_t(s, a) + \sum_{x \in \mathcal{X}^K} P(x|s, a)\langle c_t(x), w_t^K(x)\rangle\right)(p_t(s, a) - p(s, a)), \tag{41}$$

which exactly matches the cumulative cost $\tilde{c}_t^{K-1}(s, x)$ computed by Algorithm 6.

The subsequent analysis is the same as the proof for Proposition 5.1, which shows that the proposition also holds for MDPs with $L = K + 1$. By induction, the proposition is proved. $\square$

PROOF FOR PROPOSITION C.2

Similar to the proof for Proposition 5.2, let us prove the following result by induction:

$$\|\tilde{M}_t^k(x) - \tilde{c}_t^k(x)\|_\infty \le (L-k)\epsilon^l + 2\eta \sum_{m=1}^{L-k-1} m^2, \quad l \le k \le L-1, \forall x \in \mathcal{X}^k. \tag{42}$$

First, in the case when $k = L - 1$, the proposition immediately holds. Assume the proposition holds when $k = K + 1$ ($l \le K \le L - 2$), that is $|\tilde{M}_t^{K+1}(x,a) - \tilde{c}_t^{K+1}(x,a)| \le (L - K - 1)\epsilon^l + 2\eta \sum_{m=1}^{L-K-2} m^2$ holds for all $(x,a) \in \mathcal{U}^{K+1}$.

Then, for $x \in \mathcal{X}^K$, recall that

$$\tilde{c}_t^K(x,a) = c_t(x,a) + \sum_{s \in \mathcal{X}^{K+1}} P(s|x,a)\langle \tilde{c}_t^{K+1}(s), w_t^{K+1}(s) \rangle, \tag{43}$$

$$\tilde{M}_t^K(x,a) = M_t^l(x,a) + \sum_{s \in \mathcal{X}^{K+1}} P(s|x,a)\langle \tilde{M}_t^{K+1}(s), g_t^{K+1}(s) \rangle, \tag{44}$$

which yields the following results for $x \in \mathcal{X}^K$ and $a \in \mathcal{A}(x)$:

$$|\tilde{M}_t^K(x,a) - \tilde{c}_t^K(x,a)|$$

$$\le \epsilon^l + \sum_{s \in \mathcal{X}^{K+1}} P(s|x,a)\left[\sum_{b \in \mathcal{A}(s)} w_t^{K+1}(s,b)\left|\tilde{M}_t^{K+1}(s,b) - \tilde{c}_t^{K+1}(s,b)\right|\right] +$$

$$+ \sum_{s \in \mathcal{X}^{K+1}} P(s|x,a)\left[\sum_{b \in \mathcal{A}(s)} \tilde{M}_t^{K+1}(s,b)\left|g_t^{K+1}(s,b) - w_t^{K+1}(s,b)\right|\right] \tag{45}$$

$$\le (L-k)\epsilon^l + 2\eta \sum_{m=1}^{L-K-2} m^2 + \sum_{s \in \mathcal{X}^{K+1}} P(s|x,a)\|\tilde{M}_t^{K+1}(s)\|_\infty \|g_t^{K+1}(s) - w_t^{K+1}(s)\|_1,$$

where the last inequality is due to Hölder's inequality. As in the proof for Proposition 5.2, we have:

$$2\eta\|\tilde{M}_t^{K+1}(s)\|_\infty \ge \|g_t^{K+1}(s) - w_t^{K+1}(s)\|_1. \tag{46}$$

Hence, the prediction error is:

$$|\tilde{M}_t^K(x,s) - \tilde{c}_t^K(x,s)| \le (L-k)\epsilon^l + 2\eta \sum_{m=1}^{L-K-2} m^2 + 2\eta \sum_{s \in \mathcal{X}^{K+1}} P(s|x,a)\|\tilde{M}_t^{K+1}(s)\|_\infty^2$$

$$\le (L-k)\epsilon^l + 2\eta \sum_{m=1}^{L-K-2} m^2 + 2\eta(L - K - 1)^2, \tag{47}$$

where the last inequality is due to the upper bound of the cumulative predictions. Therefore, the proposition holds for layer $K$. By induction, the proposition is proved. $\square$

