# OpenReview forum: "Learning with Real-time Improving Predictions in Online MDPs"
_ICLR.cc/2025/Conference — Submitted to ICLR 2025_

### Official Review · Reviewer_sRrC · 2024-10-27

**Soundness:** 3
**Presentation:** 3
**Contribution:** 3
**Rating:** 5
**Confidence:** 3

**Summary:**

This paper introduce the Decoupling Optimistic Online Mirror Descent (DOOMD) algorithm, a novel approach for episodic Markov Decision Processes that incorporates real-time updates in predictions. Unlike traditional methods with fixed policies per episode, DOOMD continuously adjusts both predictions and policies. By decomposing decision-making across states, each state executes a unique sub-algorithm that considers immediate and future decision impacts. This paper also establish a sub-linear regret bound for DOOMD, ensuring a worst-case performance guarantee.

**Strengths:**

- While computing different policies across episodes or steps is an established concept, this paper introduces a novel approach by computing an optimal policy based on distinct subspaces within the state space. Unlike prior decision-making algorithms in non-stationary or time-varying environments that focus on optimal policy computation over the time dimension, this work innovates by emphasizing optimal policy computation across the spatial dimension of the state space.

**Weaknesses:**

Thanks for the paper and the efforts. Please see the *questions* for the weakness.

**Questions:**

- The reviewer understands that the primary algorithm divides the state space into subspaces and computes policies independently for each. Could the authors clarify how, in practical terms, one might decide on these partitions?

- On line 38: It appears that the route $1 \to 2 \to 3$ is more optimal than $1 \to 2 \to 4$ for Figure 1-(a). Please verify this.

- On line 56: Could you please be specific on how the current paper's approaches can help the development of Large Language Models?

- Please consider referring to the following papers that use predictions to dynamically update policies (or relate to optimal early stopping for policy updates):

  - Lee, H., Jin, M., Lavaei, J., & Sojoudi, S. Pausing Policy Learning in Non-stationary Reinforcement Learning. Forty-first International Conference on Machine Learning.
  - Pettet, G., Mukhopadhyay, A., & Dubey, A. (2022). Decision Making in Non-stationary Environments with Policy-Augmented Monte Carlo Tree Search. arXiv preprint arXiv:2202.13003.

- Regarding the partition $\mathcal{X} = \bigcup_{l \in \mathcal{L}} \mathcal{X}^l$, are $\mathcal{X}^l, l \in [L]$ disjoint sets?

- Lemma 3.1 would benefit from further elaboration. Could the authors clarify the significance of Lemma 3.1 and its implications?

- On line 190: This equation seems to need adjustment. Here, $M^l_t$ is defined as the prediction cost from $l$ to $L-1$, with the cost’s input as $\mathcal{U}$. Should there be a new notation, such as $c^l_t$, representing the actual cost function from $\mathcal{U}^l_t \to [0,1]$?

- With this in mind, Assumption 3.2 appears somewhat straightforward, as $c_t(u)$ and $M^l_t$ account for different input lengths.

---

> ### Author Response · Authors · 2024-11-21
> **Response**
>
> We appreciate the reviewer’s valuable feedback on our work. Below is our itemized response to the reviewer’s comments.
>
> 1. **State space decomposition**: We thank the reviewer for this insightful question. There are two levels of decomposition involved in our work, as explained below:
>     - Grouping states into layers: The algorithm is designed for loop-free episodic MDPs, which require the state space to be partitioned into layers. This partitioning reflects the inherent relationships between states. For instance, in the five-state example in Figure 2, the loop-free structure naturally results in three layers. If the MDP does not initially have a loop-free structure, it must first be transformed into an equivalent layered structure, as described in Section 3.1.
>     - Decomposing decisions: A key innovation in our proposed algorithm is the decomposition of decisions across all states. Each state independently implements a sub-algorithm to control its contribution to the total regret. For this step, we are decomposing the decisions to all the states, rather than any subset of the state space.
>
> 2. **Optimal route**: Thanks for the question. We mentioned in the manuscript that the traveler is moving from Node 1 to 4. Based on the initial predictions provided in Figure 1(a), the optimal route is Node 1 → Node 2 → Node 4.
>
> 3. **Development of predictors**: We appreciate the reviewer’s question and apologize for any confusion caused in the previous manuscript. Our claim is that with the development of Large Language Models (LLMs), there is growing interest in utilizing these models as "predictors," making decision-making under predictions increasingly relevant.
>
>     However, we emphasize that this paper does not aim to optimize or improve the predictors themselves. Instead, we assume the predictor is exogenous and fixed and focus on optimizing decision-making given these predictions.
>
> 4. **Relevant papers**: Thank you for pointing out relevant literature. We have reviewed and cited these papers in the revised manuscript to provide appropriate context.
>
> 5. **Disjoint sets**: Thanks for the question. Yes, they are disjoint sets. This is also fundamentally due to the requirement for the loop-free structure, as also mentioned in response to Comment (1).
>
> 6. **Lemma 3.1**: We appreciate the reviewer’s question and apologize for any confusion caused in the previous manuscript. As discussed earlier, the proposed algorithm is designed for loop-free episodic MDPs, which require the state space to be partitioned into layers.
>
>     Through Lemma 3.1, we aim to clarify that this requirement is not restrictive and does not limit the applicability of our algorithm. Any episodic MDP, even if not initially layered, can be transformed into a layered structure where the algorithm can still be applied.
>
>     To accommodate the newly added numerical examples, we had to remove the explicit lemma and incorporate its content into the main text due to space constraints.
>
> 7. **Notations on the cost function**: We appreciate the reviewer’s careful observation and apologize for the confusion caused in the manuscript. Below, we provide both an intuitive and mathematical explanation for the notation:
>
>     - Intuitive explanation: The superscript $l$ in the prediction $M_t^l$ indicates that the predictions are dynamically updated for each layer, and this prediction is received on layer $l$. Conversely, the cost function $c_t$ refers to the true but unknown costs in episode $t$, which does not vary with the layer $l$. As a result, it does not require a superscript.
>
>     - Mathematical explanation: as also pointed out by the reviewer, $M_t^l$ is defined for state-action pairs from layer $l$ to layer $L-1$. For $u∈U^k$ and $k≥l$, this state-action pair $u$ belongs to the definition domain. Meanwhile, the cost function $c_t$ is defined for the entire state-action pairs $U$, which also includes $u$. Therefore, the notation is mathematically rigorous.
>
> 8. **Comparability of costs and predictions**: Thanks for the question, and we again apologize for the confusion caused in the manuscript. As clarified in response to Comment (7), there is no error in the notations. For $u∈U^k$ and $k≥l$, both $c_t (u)$ and $M_t^l (u)$ are scalars, and hence directly comparable.
>
>     The intuition behind this assumption is that we want to leverage updated predictions only when we believe that they are more accurate than previous ones. Including less accurate predictions would contaminate our knowledge of the system, making updates counterproductive.
>
>     Again, to accommodate the newly added numerical examples, we had to remove the explicit assumption and incorporate its content into the main text due to space constraints.

---

### Official Review · Reviewer_utSo · 2024-11-03

**Soundness:** 3
**Presentation:** 4
**Contribution:** 2
**Rating:** 5
**Confidence:** 4

**Summary:**

This paper introduces a new framework that enables a more general and flexible interaction between the learner and the environment in online episodic MDPs. Different from traditional methods that use a fixed policy throughout each episode, this framework allows for updates to both predictions and policies within an episode.

The authors introduce the concept of cumulative cost, which considers both immediate costs and the long-term effects on future decisions. Building on this idea, they propose the Decoupling Optimistic Online Mirror Descent (DOOMD) algorithm.

A \sqrt{T} regret bound is established in this work.

**Strengths:**

This paper is straightforward and easy to follow.

It introduces a new framework that facilitates a more general and flexible interaction between the learner and the environment in online episodic MDPs.

**Weaknesses:**

This paper appears to consider only the deterministic transition case, though it mentions that the approach can be easily generalized to the stochastic case. Could you provide more details on how this generalization would work?

The paper lacks technical novelty, as many of the proofs are largely adapted from previous works such as Zimin & Neu (2013).

It seems that this paper assumes the agent has access to the cumulative cost, which is a stronger assumption. However, it’s unclear if this assumption actually leads to an improved regret bound.

Related to the above, it's not clear how the regret bound of DOOMD compares to those in previous studies. Could you include further discussion on the sharpness of the results, e.g., in terms of factors like ∣S∣ and ∣A ∣? Without stronger guarantees, it’s hard to see the why allowing changing the policy within the episode is favorable, especially since it doesn’t achieve a better regret bound.

**Questions:**

This paper addresses the transition to a from a nonlayered structure to a layered structure. However, the sharp bounds in layered Markov Decision Processes (MDPs) do not appear to be easily transferable to unlayered MDPs and vice versa. A straightforward conversion on a bound between these two settings could result in a more relaxed dependence on H.

It appears that the paper assumes the learner receives an updated cost prediction for state-action pairs across all subsequent layers. Can you provide a motivating example to justify this assumption in real-world scenarios, particularly when dealing with a stochastic transition function? Additional elaboration on this would be helpful.

The main text lacks self-containment. It would be beneficial to discuss some high-level concepts of Algorithms 3 to 5 within the main text.

---

> ### Author Response · Authors · 2024-11-21
> **Response Part 1**
>
> We sincerely thank the reviewer for their valuable feedback and constructive comments on our work. Below is our itemized response to the reviewer’s comments.
>
> ### **Response to weaknesses**
> 1. **Generalization to stochastic transitions**: Thank you for highlighting this point. We apologize for any confusion caused in the previous manuscript. In the revised version, we have extended the proposed method to stochastic transitions, as detailed in Appendix C.
>
>     Specifically, with stochastic transitions, the regret can still be decomposed into two components
>     - One component arises directly from decisions at the current layer.
>     - The other component stems from sub-optimal decisions on previous state-action pairs
>
>     By slightly modifying the algorithm, we demonstrate that it is still possible to control the regret effectively.
>
> 2. **Technical novelty**: We appreciate the reviewer’s comment, but we respectfully disagree with it. While our work builds on ideas from the Optimistic Online Mirror Descent (OOMD) algorithm (Rakhlin & Sridharan, 2013) for handling predictable cost sequences, it addresses a fundamentally different problem. Unlike conventional online learning algorithms such as OOMD, which rely on static predictions that remain unchanged during an episode, our framework involves dynamically updated predictions and policies.
>
>     In fact, our scenario involves two levels of dynamics:
>     - Between-episode dynamics: Cost functions vary across episodes.
>     - Within-episode dynamics: Predictions and decisions are updated during an episode.
>
>     The conventional online learning and online optimization mentioned in our paper, including the OOMD algorithm, primarily address between-episode dynamics. In contrast, they overlook the within-episode dynamics that are central to our formulation.
>
>     To handle these complexities, we developed the following key insights:
>     - Dynamic Decomposition: We rigorously decompose the dynamically updated decisions, allowing us to analyze the contribution of each decision.
>     - Regret Component Control: We devised a method to subtly decompose the total algorithmic regret across all decisions, ensuring that each regret component remains effectively controllable.
>
>     These innovations allow the algorithm to handle both levels of dynamics, distinguishing our approach from existing methods.
>
> 3. **Performance comparison**: We appreciate the reviewer’s insightful comment regarding the performance comparisons.
>
>     We would like to first clarify that our algorithm does not make any assumptions beyond full-information feedback, which is a standard assumption adopted by many previous works. The cumulative costs are computed based on known costs for state-action pairs.
>
>     In addition, in the revised manuscript, we include a numerical example using a routing scenario based on real-world data. For comparison, we use the original OOMD algorithm as one of the benchmarks. This algorithm utilizes predictions only at the beginning of each episode without further updates. Besides, we consider two distinct scenarios in our simulations:
>     - Naturalistic: Predictions are derived from actual data.
>     - Adversarial: Predictions are intentionally contaminated to simulate extreme cases.
>
>     The results demonstrate the following:
>     - In the naturalistic scenario, our algorithm performs better than the benchmarks when a suitable learning rate is selected.
>     - In the adversarial scenario, our algorithm significantly outperforms the benchmarks, showcasing its robustness.

---

> ### Author Response · Authors · 2024-11-21
> **Response Part 2**
>
> ### **Response to questions**
>
> 1. **Effects of transformation**: We agree with the reviewer’s observation that transforming a non-layered episodic MDP into a layered structure could potentially impact the regret bound.
>
>     However, this issue is common to online learning algorithms designed for loop-free MDPs, as seen in works such as Ghasemi et al., 2021; Rosenberg & Mansour, 2019, just to name a few. Importantly, we can prove that the transformation enlarges the state space by at most the size of the choice set in the original non-layered episodic MDP. For instance, even in scenarios with complex road networks, practical choice sets (e.g., the 3-5 routes typically recommended by Google Maps) tend to be small. Consequently, the regret bound is not significantly affected in such cases.
>
>     If the reviewer finds it necessary, we would be happy to include a formal proof of this proposition in the revised paper.
>
> 2. **Examples of updated predictions**: Thank you for this insightful question. Let us provide examples for both deterministic and stochastic transitions:
>
>     - Deterministic Transitions: Consider a routing scenario where travel time predictions are available for all links (e.g., distinct from Google Maps' overall path predictions). Initially, when the agent departs from the origin, they receive travel time predictions. However, as traffic conditions change during the journey, predictions are updated at subsequent intersections to reflect new information.
>
>     - Stochastic Transitions: Let us keep the original routing scenario and the travel time predictions, but now suppose we are a navigation service provider offering route suggestions to an end user. The end user may follow our suggestions with a certain probability or choose an alternative route. This introduces stochasticity in state transitions.
>
> 3. **Self-containment of main text**: We appreciate the reviewer’s comments and apologize for any confusion caused in the previous manuscript. To address this, we have enhanced the explanation of the algorithm in the revised version, ensuring it is conceptually clear without requiring reference to the appendix. Furthermore, Section 4.1 is devoted to illustrating the algorithm with a simple example, while Section 4.2 presents a detailed generalization of the method. We believe this approach strikes a balance between clarity and space limitations.
>
> **Reference**
>
> Ghasemi, M., Hashemi, A., Vikalo, H., & Topcu, U. (2021). Online Learning with Implicit Exploration in Episodic Markov Decision Processes. Proceedings of the American Control Conference, 2021-May, 1953–1958. https://doi.org/10.23919/ACC50511.2021.9483085
>
> Rakhlin, A., & Sridharan, K. (2013). Online learning with predictable sequences. Journal of Machine Learning Research, 30, 993–1019.
>
> Rosenberg, A., & Mansour, Y. (2019). Online stochastic shortest path with bandit feedback and unknown transition function. Advances in Neural Information Processing Systems, 32.

---

### Official Review · Reviewer_Rord · 2024-11-04

**Soundness:** 3
**Presentation:** 2
**Contribution:** 3
**Rating:** 5
**Confidence:** 4

**Summary:**

The paper proposes an algorithm to solve episodic MDP with deterministic transitions by allowing the policies to update continuously within an episode. To this end, the paper builds on Optimistic mirror descent (OMD), which provides a prediction functionality via the so-called predictable sequences.

**Strengths:**

The model is novel and exciting; while the broad area of improving RL with predictions is not new, the methodology and the model are new.
The methodology nicely adopts the optimistic mirror descent technique for solving deterministic episodic MDP with clear and complete regret analysis.

**Weaknesses:**

1) I have significant concerns about the clarity of the content presentation and layout, e.g.,
Algorithm 2 is incomprehensible without looking at the appendix, which does not seem to be a good workaround around the page limit of the submission at the cost of clarity.

2) To my understanding, in episodic MDPs, the learned policy is itself non-stationary, I.e., it depends on h. The policy takes into account how many times steps are remaining in the episode. While the setting is different here it is not clear what is the baseline case that the paper is trying to contrast. Can something easier/computationally faster be done when there are no predictions, what will be the regret then?

3) The nature of predictions is not clearly defined in the introduction. The authors need to consider a comparison with a large body of work with online learning with predictions (which is not done) e.g. https://proceedings.mlr.press/v119/bhaskara20a/bhaskara20a.pdf (Online Learning with Imperfect Hints) and related papers cited and citing the linked paper.

4)Lines 181-182 say the methodology can be generalized to stochastic transition. How?

5)The model has unbounded robustness. Ideally, it is expected that the model should be robust when errors in the predictions are really high. What can be done to mitigate this?

6)While the paper builds on the ideas of Optimistic mirror descent for predictable loss sequences, it does not explain the key technical insights that make the algorithm applicable to this setting. This is an important constituent of a well-written theory paper.

**Questions:**

Please refer to the above section

---

> ### Author Response · Authors · 2024-11-21
> **Response Part 1**
>
> We appreciate the reviewer’s valuable feedback on our work. Below is our itemized response to the reviewer’s comments.
>
> 1. **Clarity of the presentation**: We appreciate the reviewer’s comments regarding the clarity of our algorithm’s presentation and apologize for any confusion caused in the previous manuscript. To address this, we have enhanced the explanation of the algorithm in the revised version, ensuring it is conceptually clear without requiring reference to the appendix.
>
>     In addition, we would like to emphasize that to enhance clarity, our paper devoted the entire Section 4.1. to illustrate the algorithm using a simple example. The algorithm in Section 4.2 is a detailed generalization of the method. We believe this is a clearer approach compared with directly presenting all the algorithm components.
>
> 2. **Stationarity and benchmarks**: We agree with the reviewers that our algorithm is also non-stationary. A short answer regarding the benchmark is that our algorithm should be compared with algorithms that rely less on predictions.
>
>     To be more specific, our scenario involves two levels of dynamics:
>     - Between-episode dynamics: Cost functions vary across episodes.
>     - Within-episode dynamics: Predictions and decisions are updated during an episode.
>
>     Therefore, the learned policy depends not only on the current episode but also on the current layer within the episode. Previous online learning and online optimization algorithms mentioned in our paper primarily address between-episode dynamics, overlooking within-episode updates. Therefore, to the best of our knowledge, no prior work has addressed both levels of dynamics simultaneously, making it challenging to identify a direct counterpart for comparison. Thus, we chose benchmarks that utilize less predictive information. For instance, conventional algorithms such as optimistic online mirror descent (Rakhlin & Sridharan, 2013) treat each episode as a whole and utilize predictions only at the start of the episode, without further updates. To strengthen our evaluation, we compare our algorithm against these benchmarks in the newly added numerical experiments in the revised manuscript.
>
> 3. **Nature of predictions**: We thank the reviewer for pointing out related literature, which we have cited in the revised manuscript. We apologize for the confusion in our previous version. The key distinction lies in the prediction models themselves: the models referenced by the reviewer are not updated within each episode, whereas ours are.
>
>     As mentioned in response to Comment (2), our scenario involves two levels of dynamics – between-episode and within-episode dynamics, whereas the models in the cited works focus exclusively on the former. This distinction is central to our formulation and highlights the novelty of our approach.
>
> 4. **Generalization to stochastic transitions**: Thank you for highlighting this point. We apologize for any confusion caused in the previous manuscript. In the revised version, we have extended the proposed method to stochastic transitions, as detailed in Appendix C.
>
>     Specifically, with stochastic transitions, the regret can still be decomposed into two components.
>     - One component arises directly from decisions at the current layer.
>     - The other component stems from sub-optimal decisions on previous state-action pairs
>
>     By slightly modifying the algorithm, we demonstrate that it is still possible to control the regret effectively.
>
> 5. **Unbounded robustness with inaccurate predictions**: Thanks for this insightful question. Strictly speaking, the regret is unbounded and grows at the rate of $O(\sqrt{T})$ even with small prediction errors. This is a common characteristic of online learning algorithms in adversarial MDPs, where the learner has no knowledge of arbitrarily changing future costs. Therefore, maintaining a bounded regret $R_T$, or achieving Hannan consistency, namely $R_T\to 0$ as $T \to \infty$, is typically infeasible.
>
>     Instead, the standard objective is to maintain a sublinear regret bound, such that the average regret $R_T/T \to 0$ as $T \to \infty$. Note that even in scenarios with extremely inaccurate predictions, as the costs are bounded between 0 and 1, the regret remains the same order $O(\sqrt{T})$. This order of regret, while unavoidable in adversarial settings, is not a serious limitation of the proposed algorithm.

---

> ### Author Response · Authors · 2024-11-21
> **Response Part 2**
>
> 6. **Technical insights**: We appreciate the suggestion to enhance the explanation of technical insights, which we have stressed in Section 4.1 of the revised manuscript.
>
>     While our work builds on ideas from the Optimistic Online Mirror Descent (OOMD) algorithm (Rakhlin & Sridharan, 2013) of handling predictable cost sequences, it addresses a fundamentally different problem. Conventional online learning algorithms like OOMD rely on static predictions that do not change within an episode. In contrast, our framework involves dynamically updated predictions and policies.
>
>     To handle these complexities, we developed the following key insights:
>     - Dynamic Decomposition: We rigorously decompose the dynamically updated decisions, allowing us to analyze the contribution of each decision.
>     - Regret Component Control: We devised a method to subtly decompose the total algorithmic regret across all decisions, ensuring that each regret component remains effectively controllable.
>
>     These technical innovations enable the algorithm to handle both levels of dynamics, setting it apart from existing methods.
>
> **Reference**
>
> Rakhlin, A., & Sridharan, K. (2013). Online learning with predictable sequences. Journal of Machine Learning Research, 30, 993–1019.

---

### Official Review · Reviewer_Ean4 · 2024-11-06

**Soundness:** 3
**Presentation:** 3
**Contribution:** 3
**Rating:** 6
**Confidence:** 3

**Summary:**

This paper studies online episodic MDP with time-varying cost functions and predictions. A novel algorithm, Decoupling Optimistic Online Mirror Descent (DOOMD), is proposed to update both predictions and policies throughout the episodes. A sublinear regret guarantee is also established to demonstrate the effectiveness of the proposed algorithm.

**Strengths:**

1. Involving real-time predictions in online MDP is an interesting idea since many real-world applications have certain predictions on the future costs.

2. The paper is well-written. The algorithm procedures are well explained.

3. The proposed algorithm can update its predictions and policies during the episode instead of at the end of each episode, which has some practical appeal.

**Weaknesses:**

1. One of my major concerns is about the assumptions. This paper considers a deterministic transition function but claims that it can be easily generalized to stochastic transitions. Can the authors provide more details on this generalization?

2. The lack of simulation results is another major weakness of this paper. The authors should provide some numerical justifications of their algorithm, hopefully in both deterministic cases and stochastic cases.

3. It is true that any episodic MDP can be transformed into a loop-free MDP. However, this comes at the cost of enlarging the state space. How does this transformation affect the regret bounds' dependence on dimensionality and episode length?

**Questions:**

There are several other papers considering predictions in online learning and online control, such as [C1] [C2].

Q1: How does the prediction model compare with the prediction models considered in [C1] and [C2]?

Q2: Besides, can the regret analysis in this paper be generalized to the prediction model in [C1] and [C2]?



[C1] Li, Y., Chen, X. and Li, N., 2019. Online optimal control with linear dynamics and predictions: Algorithms and regret analysis. Advances in Neural Information Processing Systems, 32.

[C2] Li, Y. and Li, N., 2020. Leveraging predictions in smoothed online convex optimization via gradient-based algorithms. Advances in Neural Information Processing Systems, 33, pp.14520-14531.

---

> ### Author Response · Authors · 2024-11-21
> **Response Part 1**
>
> We appreciate the reviewer’s insightful comments on our work. Below is our itemized response to the reviewer’s comments.
>
> ### **Response to weaknesses**
> 1. **Generalization to stochastic transitions** : Thank you for highlighting this point. We apologize for any confusion caused in the previous manuscript. In the revised version, we have extended the proposed method to stochastic transitions, as detailed in Appendix C.
>
>    Specifically, with stochastic transitions, the regret can still be decomposed into two components
>     - One component arises directly from decisions at the current layer.
>     - The other component stems from sub-optimal decisions on previous state-action pairs
>
>    By slightly modifying the algorithm, we demonstrate that it is still possible to control the regret effectively.
>
> 2. **Simulation results**: We acknowledge the absence of numerical results in the previous version and have addressed this concern in the revised manuscript. We now include a numerical example based on a routing scenario using the real-world METR-LA dataset.
>
>     As regret bounds are primarily designed to ensure worst-case performance, it is inherently challenging to evaluate them directly using real-world data, as such data may not reflect the "worst-case" conditions. To address this, we consider two distinct scenarios in our simulations:
>     - Naturalistic: Predictions are derived from actual data.
>     - Adversarial: Predictions are intentionally contaminated to simulate extreme cases.
>
>     The results demonstrate the following:
>     - In the naturalistic scenario, our algorithm performs better than the benchmarks when a suitable learning rate is selected.
>     - In the adversarial scenario, our algorithm significantly outperforms the benchmarks, showcasing its robustness.
>
> 3. **Effects of transformation**: We agree with the reviewer’s observation that transforming a non-layered episodic MDP into a layered structure enlarges the state space, which could potentially impact the regret bound.
>
>     However, this issue is common to online learning algorithms designed for loop-free MDPs, as seen in works such as Ghasemi et al., 2021; Rosenberg & Mansour, 2019, just to name a few. Importantly, we can prove that the transformation enlarges the state space by at most the size of the choice set in the original non-layered episodic MDP. For instance, even in scenarios with complex road networks, practical choice sets (e.g., the 3-5 routes typically recommended by Google Maps) tend to be small. Consequently, the regret bound is not significantly affected in such cases.
>
>     If the reviewer finds it necessary, we would be happy to include a formal proof of this proposition in the revised paper.

---

> ### Author Response · Authors · 2024-11-21
> **Response Part 2**
>
> ### **Response to questions**
> 1. **Difference of prediction models**: We appreciate the reviewer for highlighting related literature, which is cited in the revised manuscript. The key distinction lies in the prediction models themselves: the models referenced by the reviewer are not updated within each episode, whereas ours are.
>
>     Specifically, our scenario involves two levels of dynamics:
>     - Between-episode dynamics: Cost functions vary across episodes.
>     - Within-episode dynamics: Predictions and decisions are updated during an episode.
>
>     The prediction models discussed in the cited works, as well as those in the context of online learning and online optimization mentioned in our paper, primarily address between-episode dynamics. In contrast, they overlook the within-episode dynamics that are central to our formulation.
>
> 2. **Generalization to other prediction models**: Thank you for raising this point. A short answer is yes, our method can deal with these prediction models but may require additional efforts.
>
>     As discussed in the paper, many conventional online learning algorithms in episodic MDPs operate with static policies updated only at the start of each episode (e.g., Fei et al., 2020; Rakhlin & Sridharan, 2013; etc.). This framework represents a degenerate case of our proposed algorithm.
>
>     However, we note that the papers cited by the reviewer involve MDP structures with nuanced features, such as switching costs or control structures. These features establish connections between episodes, which is not directly reflected in the MDP formulation of our work. Extending our algorithm to address such scenarios may require additional effort and adjustments to the underlying framework.
>
> **Reference**
>
> Fei, Y., Yang, Z., Wang, Z., & Xie, Q. (2020). Dynamic regret of policy optimization in non-stationary environments. Advances in Neural Information Processing Systems, 2020-Decem.
>
> Ghasemi, M., Hashemi, A., Vikalo, H., & Topcu, U. (2021). Online Learning with Implicit Exploration in Episodic Markov Decision Processes. Proceedings of the American Control Conference, 2021-May, 1953–1958. https://doi.org/10.23919/ACC50511.2021.9483085
>
> Rakhlin, A., & Sridharan, K. (2013). Online learning with predictable sequences. Journal of Machine Learning Research, 30, 993–1019.
>
> Rosenberg, A., & Mansour, Y. (2019). Online stochastic shortest path with bandit feedback and unknown transition function. Advances in Neural Information Processing Systems, 32.

---

> > ### Comment · Reviewer_Ean4 · 2024-11-22
> >
> > Thanks for the detailed responses, which have addressed my concerns satisfactorily. I have increased my score.

---

### Meta-Review · Area_Chair_wtJB · 2024-12-21

**Metareview:**

Summary:
This paper investigates online episodic Markov Decision Processes (MDPs) with time-varying cost functions and predictions. The authors propose a novel algorithm, Decoupling Optimistic Online Mirror Descent (DOOMD), which updates both predictions and policies across episodes. They also establish a sublinear regret guarantee to demonstrate the algorithm's effectiveness.

Strengths:
The proof is technically rigorous, and the presentation is clear.

Weaknesses:
As noted by other reviewers, the scope of this paper is too limited. The initial version of this work only addresses deterministic and known transitions. Although the authors later included a discussion on stochastic known transitions, this remains far from comprehensive. Given the significant body of existing research on stochastic unknown transitions, this omission is a clear drawback. Furthermore, the significance of studying time-varying cost functions has not been convincingly demonstrated.

Decision:
Reject.

**Additional Comments On Reviewer Discussion:**

The initial version focuses solely on deterministic and known transitions. While the authors have since added a discussion on stochastic known transitions during discussion, the treatment remains incomplete. Considering the extensive body of work on stochastic unknown transitions, this limitation is a significant drawback. Additionally, the practical importance of studying time-varying cost functions has not been clearly established.

---

### Decision · Program_Chairs · 2025-01-22

Reject